# Anxious about rejection, avoidant of neglect: Infant marmosets tune their attachment based on individual caregiver's parenting style

Saori Yano-Nashimoto [1,2,15], Anna Truzzi [1,3,4,15], Kazutaka Shinozuka [1,13,15], Ayako Y. Murayama[1,5,6,14], Takuma Kurachi[1,7], Keiko Moriya-Ito[8], Hironobu Tokuno[8,16], Eri Miyazawa[1], Gianluca Esposito [1,4], Hideyuki Okano [5,6], Katsuki Nakamura[9], Atsuko Saito [1,10] ✉ & Kumi O. Kuroda [1,11,12] ✉

Children's secure attachment with their primary caregivers is crucial for physical, cognitive, and emotional maturation. Yet, the causal links between specific parenting behaviors and infant attachment patterns are not fully understood. Here we report infant attachment in New World monkeys common marmosets, characterized by shared infant care among parents and older siblings and complex vocal communications. By integrating natural variations in parenting styles and subsecond-scale microanalyses of dyadic vocal and physical interactions, we demonstrate that marmoset infants signal their needs through context-dependent call use and selective approaches toward familiar caregivers. The infant attachment behaviors are tuned to each caregiver's parenting style; infants use negative calls when carried by rejecting caregivers and selectively avoid neglectful and rejecting caregivers. Family-deprived infants fail to develop such adaptive uses of attachment behaviors. With these similarities with humans, marmosets offer a promising model for investigating the biological mechanisms of attachment security.

Early life adversity affects infants' cognitive, social, and emotional development, ultimately increasing the risks of various medical conditions and premature death[1–3]. Among the factors comprising the early life environment, the stability and quality of the relationship with the primary caregiver are critical for infants' sense of security because infants are born immature and require extensive care for survival among all mammals. The experiences gained through interactions with the primary caregiver(s) (often the mother and other family members) are also essential for learning life skills and social behaviors in many species. Thus, infants have an innate motivation to seek and maintain proximity with the primary caregiver by approaching and signaling, collectively called the attachment system[4].

While the basic attachment system is innate, infants adjust their attachment pattern depending on the quantity and quality of care received. It is theorized that if the caregiver inflicts fear or is insensitive to infant distress, infant attachment becomes insecure, i.e., infants are not fully confident in the caregivers' availability and responsiveness[5–7]. However, the discerned association between quality of rearing and attachment security is not large, possibly due to difficulties in controlling for other parameters such

as genetic factors and the role of multiple caregivers in human studies[8–11]. Thus, a nonhuman animal model should be established further to dissect the developmental mechanisms of infant attachment security. For this purpose, rodent infants have been studied extensively and shown to exhibit many attachment behaviors in common with humans[2,12,13]. Still, the attachment of rat or mouse pups is not as selective to a particular individual as humans, presumably because these species may engage in communal nursing[14]. In contrast, the infant attachment of primates is shown to be more selective toward the attachment figure (usually the biological mother) and is more profoundly impacted by maternal deprivation and isolation rearing[15–18] than that of rodents[19–21]. Yet, direct and quantitative examinations of infant attachment security have been limited to a few studies (see refs. [22,23]).

Biparental or cooperative infant care in primates is limited to several family-living species, including Callitrichidae (marmosets and tamarins), *Plecturocebus* (titi monkeys), and *Aotus* (owl monkeys)[24–30]. Although these New World monkeys are genetically more distant from humans than Old World monkeys, their cooperative infant care systems and the resulting shared infant attachment present a significant interest due to their

similarities with those of humans[31,32] Among these, common marmosets (*Callithrix jacchus*) are a promising primate model with cutting-edge research resources such as genetic manipulation tools and multiple kinds of brain atlases and databases[33–35]. At one birth, two infants are generally raised and carried almost continuously during the first postnatal months. Infant carrying impedes the carrier's locomotor activities and thus is shared by the family members: the mother, father, and older siblings[36]. Our previous study[37] identified two independent (allo)parenting parameters, *sensitivity* to infant distress and *tolerance* to infant carrying, similar to parenting styles established in humans[38,39]. Furthermore, the molecularly defined subregion of the medial preoptic area in the basal forebrain specifically regulates caregivers' *tolerance* in marmosets[37]. These data suggest the common neurobiological mechanism of infant caregiving behaviors across primates.

Additionally, marmosets' vocal communication has attracted considerable attention for its complexity and human language-like features such as turn-taking and infant babbling, or continuous strings of multiple call types which can last for minutes[40–42]. Marmosets' multiple call types show distinct acoustic features and include phee, twitter, tsik, trill, chatter, and Chirp (Supplementary Table 1)[43,44]. Infants also emit an additional call type, originally termed "ngä"[43,45] and recently as "cry"[42] (we use "cry" for the final version of this manuscript, according to the request from one reviewer and the editor). Vocal development of infant marmosets has been mostly studied under isolated conditions and is reported to shift infantile cry to tonal phee calls, due to the maturation of the vocal apparatus[46,47] and social learning from vocal feedback from parents[48,49]. However, another study reported that infants possess the ability to produce phee right after birth and retain cry calls at 62 postnatal weeks[50]. Moreover, as very young infants are continuously carried by the caregiver and infant calls a function to attract parental approach[51–53], an investigation of infant call development within intact family settings should be performed (see refs. [42,54]).

Thus, this study investigates the relationship between parenting styles and infant attachment behaviors including vocal communications, utilizing the high natural variations of the parenting parameters and the experimental manipulation of rearing conditions.

## Results

### The infant retrieval assay to study caregiver–infant interactions

The families were kept in a large family cage consisting of two to three connected cubicle cages. The infant retrieval assays, or brief separation and reunion from the infants' viewpoint, were conducted utilizing two cubicle cages within their home cage. A caregiver, either a mother, father, or an older sibling of the infant, was placed in one cage, and an infant in a wire basket was placed in an adjacent cage connected via a tunnel with a shutter (Fig. 1a) (Supplementary Movie 1). After the shutter was opened, the caregiver typically entered the infant cage, approached, leaned into the basket, and came in contact with the infant. The infant climbed over the caregiver's trunk, designated as infant retrieval and the start of carrying (Fig. 1b). After retrieval, the caregiver carried the infant for varying durations and may have eventually started rejecting the infant by rolling on the floor, pushing, and biting the infant. These behaviors often caused the removal of the infant from the carrier's body (Supplementary Movie 2). The session was continued for 600 s after the first retrieval or from the assay start (see Supplementary Table 2, see the "Methods" section).

Detailed analyses of 286 infant retrieval assay sessions were conducted involving 7 families, 25 infants, and 55 infant–caregiver dyads from postnatal day (PND) 1 to 36 (Supplementary Table 2), and confirmed for sufficient inter-observational reliability with the on-site behavioral coding presented in our previous study (Supplementary Fig. 1)[37]. The caregiver's and infant's behaviors and calls listed in Supplementary Table 1 were analyzed at a subsecond scale, with 0.2-s bins using video and vocal recordings of the sessions. Based on the caregiver–infant interactions, the total period of each assay was broken down into five social contexts, which were mutually exclusive and collectively exhaustive (Fig. 1b): *Alone_BeforeRET*, when the infant was not carried yet before the first retrieval; *Holding*, when the caregiver was carrying the infant without locomotion;

*Transport*, when the caregiver was carrying the infant and locomoting; *During_Rejection*, the period from the start of rejection to 9.4 s after the end of rejection (see below and Fig. 1b legend for this definition); and *Alone_AfterRET*, when the infant was not carried after the first retrieval occurred (Supplementary Table 1).

### Caregiving parameters define parenting styles

For a data-driven analysis of the caregiver–infant dyadic relationship, we first examined the correlation matrix of all the parameters of caregivers' and infants' behaviors (Supplementary Tables 1, 3) during postnatal weeks 0–4 (Fig. 1c, Supplementary Tables 4, 5), and the candidate correlations were further analyzed. Because the parameters may or may not show normal distributions, non-parametric Spearmann's rank correlations (the left-bottom half of Fig. 1c) and parametric Pearson's product–moment correlations (the right-top half) for parameters derived from each infant retrieval assay session are shown, and we present the *p* and *r* values in the non-parametric statistical results below.

The screening analysis in Fig. 1c suggests that infant behaviors have strong correlations with the caregiver's infant-directed physical behaviors (the green rectangles of Fig. 1c). As the most fundamental (allo)parental behaviors of each caregiver[37], we define three caregivers' parameters: (i) the retrieval latency (*RET_Latency*), the time required for infant retrieval, which negatively represented the *sensitivity* of the caregiver toward infant distress vocalizations; (ii) the rejection rate (*%Rejection*), which negatively represented the *tolerance* of the caregiver to infant carrying; and (iii) the carrying rate (*%Carry*), the total carrying duration divided by the session length, which represented the total infant care quantity. *RET_Latency* and *%Rejection* were mutually independent ($r = 0.0579$, $p > 0.05$), and both contributed to net infant carrying care (*%Carry*−*RET_Latency*: $r = -0.6059$, $p < 0.001$, *%Carry*−*%Rejection*: $r = -0.5783$, $p < 0.001$). These caregiving parameters obtained in the present study were stable in each caregiver toward multiple infants across births, and consistent with their other (allo)parental indices obtained by the intact family observation in our previous study[37] (Supplementary Table 3) (e.g., *%Carry* vs. *Scan_Carrying rate*, *Family_Carrying Duration (B)*). These observations confirm the existence of a caregiver–inherent (allo)parenting style as demonstrated[37].

Along with infant development, the latencies of caregivers' approach to the infant increased (Fig. 1d), while *%Rejection* remained consistent (Fig. 1e). The net *%Carry* (Fig. 1f; $r = -0.2845$, $p = 0.0067$) and the caregiver–infant contact declined after postnatal week 4 ($r = -0.2808$, $p = 0.0094$), indicating that the parent–infant interactions were essentially stable during the first postnatal month (Fig. 1f, g). No significant differences among mothers, fathers, or older siblings were found in these core caregiving parameters (Fig. 1e, f), while male caregivers were found to exhibit more "*in basket*" (a kind of object play) and naturally, shorter periods of breastfeeding (Fig. 1c, $r = 0.2934$, $p < 0.001$ (in basket); $r = -0.2766$, $p = 0.0127$ (breastfeeding)).

Compared to the caregivers' infant-directed behaviors, caregivers' vocalizations (calls) and non-infant-directed behaviors showed less-pronounced associations with infant's behaviors and calls (Fig. 1c). Thus we focus on the caregiver's infant-directed behaviors hereafter.

### Approach components of the attachment system of infant marmosets

Infants seek and maintain the proximity of specific familiar individuals using two kinds of attachment behaviors: (1) *approach* (seeking, following, clinging) and (2) *signaling* (crying, smiling, and gestures)[4]. In the first postnatal month of marmoset infants, approach behaviors were manifested mainly by actively clinging to the caregiver's body whenever the caregiver made contact (Supplementary Movie 1) and by not breaking the contact unless the caregiver rejected them.

To examine the *selectivity of infant approach* behavior, we performed additional infant retrieval assays with the dyad of an infant and an unfamiliar adult to compare the latency of infants clinging to their parents or unfamiliar adults with multiple parental experiences (Fig. 2a–c). Infants

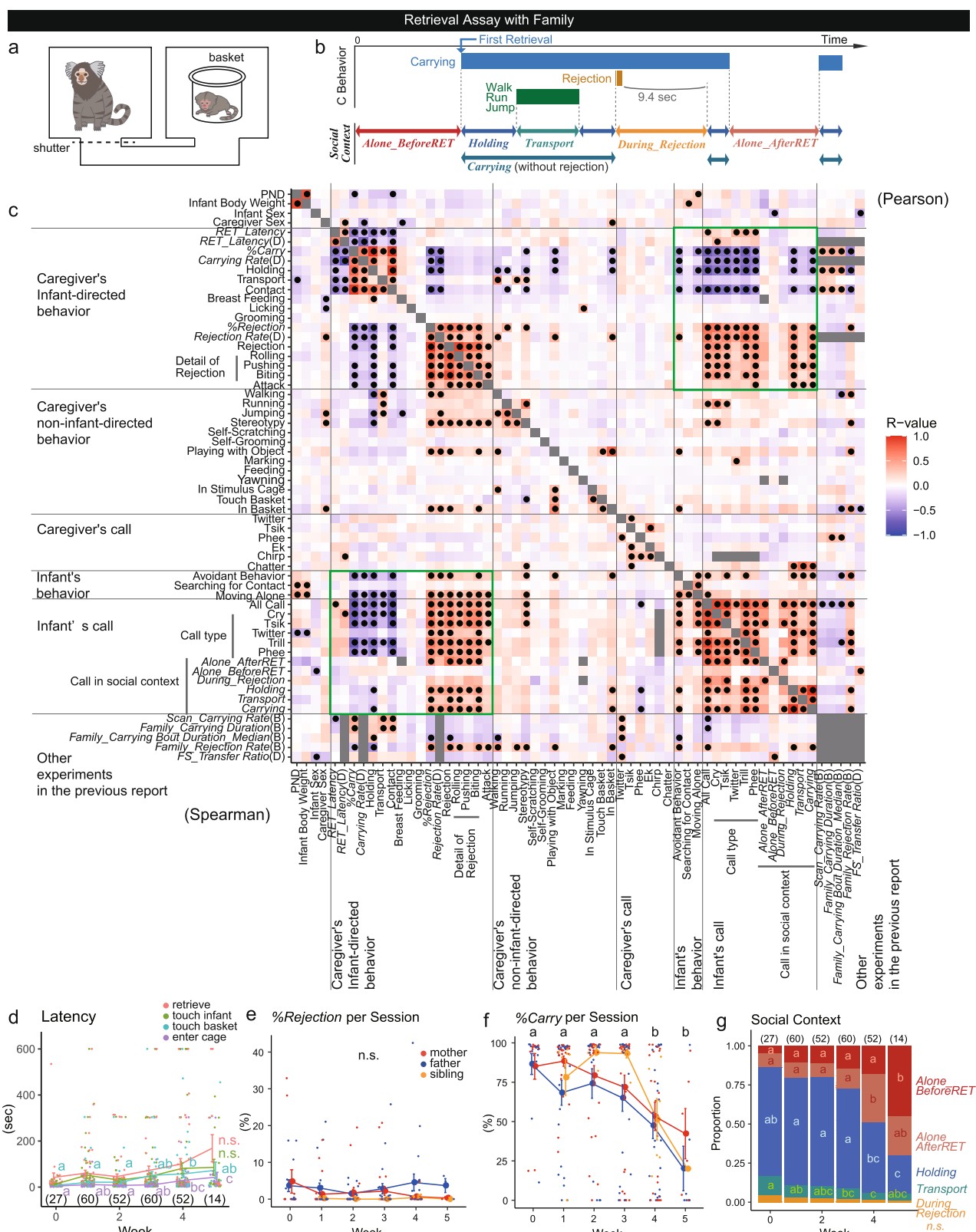

Retrieval Assay with Family

instantaneously clung to their parents but not to the unfamiliar caregivers (Fig. 2a). In contrast, parental latencies to contact with their own or unfamiliar infants were highly variable and did not reach statistical significance (Fig. 2b, the generalized linear mixed model (GLMM), $p = 0.2040$). The carrying rate, which both infants and parents could contribute to, was low for the unfamiliar parent–infant dyad (Fig. 2c). These results strongly

suggest that marmoset infants develop a selective attachment with familiar caregivers.

*Contact-breaking* behavior by infants was seen in two ways: one was when the infants were in contact with a caregiver but did not cling (passive), and the other was voluntary dismounting from the caregiver's body (active). Practically, however, it may not be always obvious whether an infant's

**Fig. 1 | Caregiver behavior in retrieval assays. a** Schematic of the infant retrieval assay in the home cage. **b** Caregiver–infant interactions and the five social contexts for the infants. Carrying includes transport and holding. A 9.4-s offset (see Fig. 2d) in *During_Rejection* was set to include the infant behaviors directly under the influence of the preceding rejection. **c** Correlation matrix of the parameters observed in the retrieval assays. The color indicates correlation coefficients (*r*-value, see Supplementary Table 4). Filled circles, *p* < 0.05 (adjusted with Holm's method, see Supplementary Table 5). The left-bottom and right-top triangular parts show Spearman's rank correlation coefficient and Pearson product–moment correlation coefficient per session, respectively. The parameters ending with either (B) or (D) are the parenting parameters derived from our previous study (Supplementary Table 3)[37], which are averaged for each dyad (D) or each birth (B) (265 sessions of 55 dyads). **d** Mean ± standard error (s.e.) latencies of the caregiver's behaviors after the

shutter's opening. Red: the first retrieval, yellow-green: the first touch of the infant, blue-green: the first touch of the basket, purple: the first reach of the infant cage. Dots show individual sessions. **e, f** Mean ± s.e. *%Rejection* (**e**) and *%Carrying* (**f**) in each postnatal week (filled circles and error bars). Red: mothers, blue: fathers, yellow: older siblings. Dots show individual sessions. **g** Proportion of duration of each social context. Red: Alone before the first retrieval, pink: alone after the first retrieval, blue: holding (carrying the infant without caregiver locomotion), green: transport (carrying the infant with caregiver walking, running, or jumping), yellow: during rejection. For **d, f**, and **g**, different letters indicate significant differences among weeks (GLMM, *p* < 0.05), and the numbers within parentheses are the numbers of the sessions (265 sessions of 55 dyads. For **e**, 238 sessions of 55 dyads; sessions without first retrieval were excluded).

dismounting was forced by caregivers' rejection or by infants' voluntary action. To empirically determine the direct influence of the preceding rejection on infant dismounting, we performed a segmented regression analysis to identify an abrupt change in the response function of a varying influential factor[55]. The result in Fig. 2d identified the inflection point at 9.4 s; thus, we defined *During_Rejection* as the period from the onset of rejection to 9.4 s after the end of rejection (Fig. 1b). The dismounting that occurred after a 9.4-s offset or those without preceding rejection were regarded as voluntary.

An infant's *avoidant behaviors* were defined as the sum of voluntary dismounting and absence of clinging when the infant was in contact with the caregiver. The frequency of avoidant behaviors per session was substantially increased after postnatal week 4 (Fig. 2e), indicating that the marmosets become autonomous. Thus, these results suggest that avoidant behaviors after postnatal week 4 are a sign of typical development, while premature (i.e., within postnatal weeks 0-3) avoidant behaviors are atypical and may be a sign of decreased infant attachment to the given caregiver (see below).

### Infant call as a signaling component of the marmoset attachment system

Next, we studied infant vocalizations in various social contexts with the caregiver, first focusing on the total amount of calls, and then on the use of distinct call types (Fig. 2f–k).

**Calls during rejection.** The infants emitted calls most frequently when they were rejected or immediately after (Fig. 2f, *During_Rejection*; Supplementary Movie 2, Supplementary Fig. 2a). In the family observation, the infant's vigorous calls during rejection appeared to attract other family members to the infant–carrier dyad and allow the infant to transfer quickly from the previous carrier to the next (Supplementary Movie 3), suggesting the signaling function of infant calls. In this way, during the first three postnatal weeks, most infants are directly transferred from one carrier to the next (see below for direct and indirect transfer of infants).

**Calls-while-not-being-carried.** Infants frequently vocalized also when they were not carried (Fig. 2f), often as a continuous string of various different call types termed babbling (Fig. 2g, h)[40,42]. We observed that these intense bouts of calling stopped immediately after the infants were carried (Fig. 2f–h). Comparisons of call frequencies at the transition of social contexts revealed that infants reduced calling when they came into contact with the caregiver with any part of the body and further withheld calling when they clung to the caregiver's body (=carrying) (Fig. 2i). These findings further support the notion that these bursts of infant calls in *Alone* contexts signaled the separation distress widely observed in mammalian infants[56] and thus were withheld immediately after contact with the caregiver.

**Ontogeny of calls in family and isolation.** The call frequencies during each social context did not change until postnatal week 5 (Fig. 2j). The increase in total calls during the test session along with infant

development (Fig. 2j) should be attributed to the significant increase in *Alone* contexts (Fig. 1g). In a separate experiment when we briefly took out these infants from the family and recorded their vocalizations in a completely isolated recording room, total call frequencies were high in the beginning and declined after postnatal week 6 (Fig. 2k, l, complete isolation) when parental carrying declined rapidly[37]. These findings altogether suggest that the calls during *Alone* contexts (calls-while-not-being-carried in the family or in complete isolation) decline during infant development due to the decline in attachment needs by infant maturation (see the "Discussion" section).

**Selective use of call types.** Infant marmosets emit various kinds of call types, including twitter, tsik, trill, phee, cry, and various combinations of these calls (Fig. 2g, h, m, Supplementary Fig. 2b, c). The ratio of infant call types depended considerably on the social context; infants emitted more trill calls when they were carried and more tsik calls when they were rejected (Fig. 2m). Twitter and cry calls were most frequent during *Alone_BeforeRET* and *Alone_AfterRET*, respectively. These call type usages did not change substantially during postnatal weeks 0–5 in the infant retrieval assays (Supplementary Fig. 2c). Thus, infant marmosets in the first postnatal month already use multiple call types selectively in each social context, although not exclusively. Moreover, infants emitted more phee calls in the completely isolated recording condition (Fig. 2l, Supplementary Fig. 3a–g) than in while-not-being-carried (*Alone*) conditions in the dyadic retrieval assays (compare Fig. 2m and Supplementary Fig. 2c). This fact implies that infants in the first week of life use phee calls as distant contact calls toward invisible family members as adults do.

Of note, in the complete isolated recording condition, the total amount of phee calls remained stable across development (Fig. 2l, Supplementary Fig. 3b). In contrast, cry and twitter calls rapidly declined during the first 8 weeks (Fig. 2l, Supplementary Fig. 3c, f), which accounted for the decrease in total calls. Thus, the previously proposed developmental increase of phee calls may be relative and caused by the developmental decline of cry and twitter calls in isolated recording conditions.

### Avoidance and anxious calls during carrying are associated with a low quantity and quality of caregiving

We next investigated the relationship between parenting styles and infant behaviors by utilizing the wide range of individual variabilities in caregivers' parameters (Fig. 3a, b, Supplementary Fig. 4a–c). As an example, male twin infants Saku (Fig. 3a) and Gaku (Supplementary Fig. 4a) had a tolerant mother, Kachan, and an intolerant, rejecting father, Tochan, who occasionally attacked his infants. The mother (Fig. 3a, Supplementary Fig. 4a) retrieved the infant quickly and carried it throughout without rejections. Both infants mostly withheld calls after retrieval and kept attached to their mothers. In contrast, the father (Fig. 3b, Supplementary Fig. 4a) repeated retrieval and rejection, resulting in fragmented carrying bouts. In the father's sessions, both infants exhibited premature avoidant behaviors (black triangles in Fig. 3b, Supplementary Fig. 4a) not only during rejection but also during carrying. Moreover, the infants did not immediately withhold calling after the father's retrieval (e.g., ~600 s in Fig. 3b). These parental behaviors

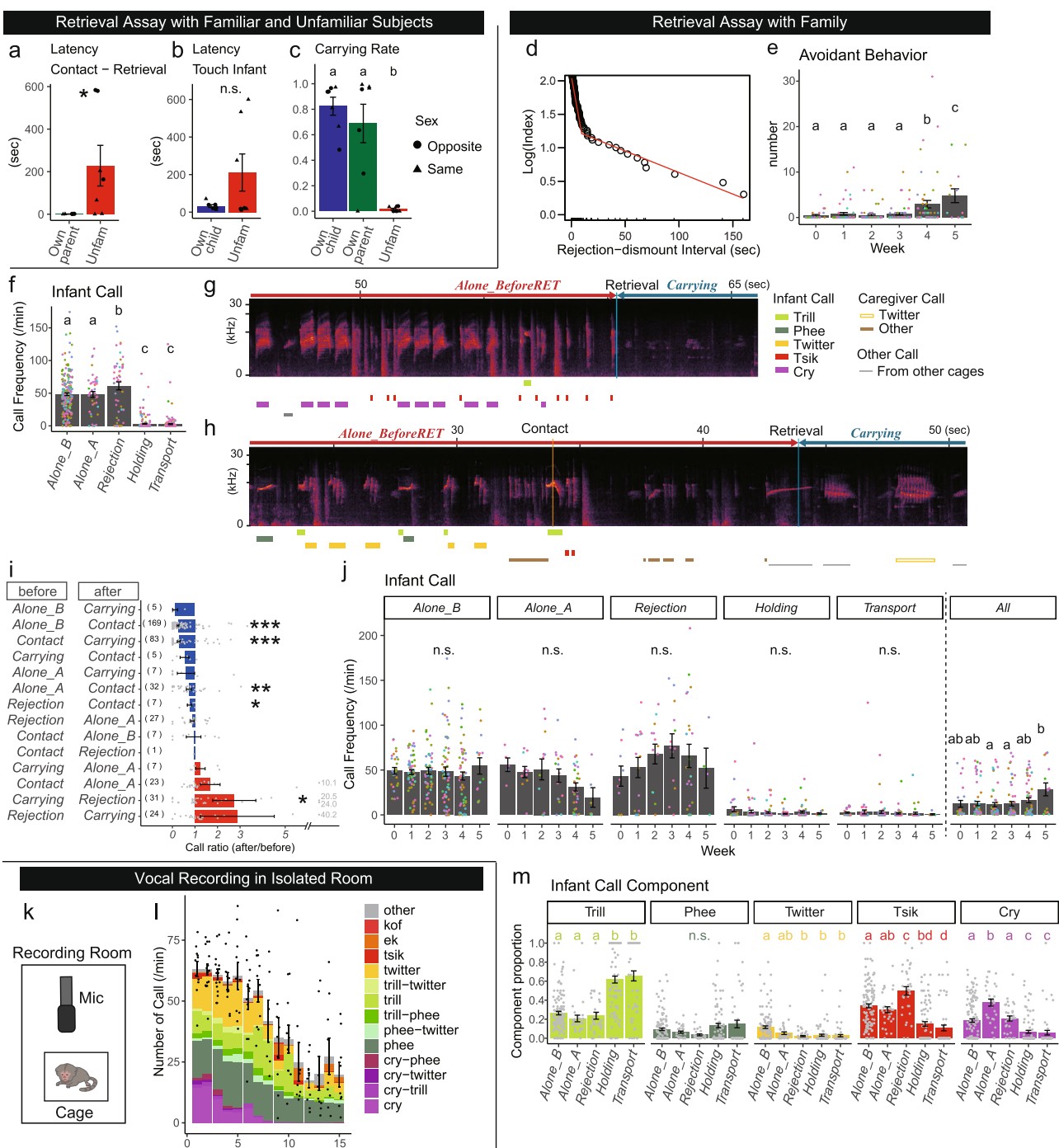

**Fig. 2 | Infant behavior in retrieval assays and isolated recordings. a–c** Mean ± s.e. of the latency of the infant retrieval after the first contact (**a**), the latency of the caregiver's first touch of the infant (**b**), the carrying rate (**c**) in the retrieval assay using unfamiliar (Unfam) or familiar caregivers (own parent or own infant) (7 sessions for each group). **d** Log-survivorship analysis of the rejection-dismount interval, defining the dismounts that occurred later than 9.4 s after the end of rejection as voluntary dismounts. Red line: the segmented regression line (164 observed dismounts). **e** Mean ± s.e. numbers of avoidant behaviors per session (265 sessions of 55 dyads). **f** Mean ± s.e. frequencies of infant calls in each social context. Alone_B: *Alone_BeforeRET*, Alone_A: *Alone_AfterRET*, Rejection: *During_Rejection* (data collected from 265 sessions of 55 dyads). **g, h** Two typical spectrograms of infant and caregiver call at the transition from *Alone_BreforeRET* to *Carrying*. After the first contact and retrieval, the infant calls immediately stopped. **i** Mean ± s.e. changes of the call frequency 10 s after/before the shift of the social contexts (i.e., ratio = 1 means no change). Wilcoxon signed-rank test with continuity

correction, *$p < 0.05$, **$p < 0.01$, ***$p < 0.001$. The numbers within parentheses are the numbers of the scenes analyzed. The gray numbers are the values of the outliers. Alone_B: *Alone_BeforeRET*, Alone_A: *Alone_AfterRET*, Rejection: *During_Rejection* (data collected from 265 sessions of 55 dyads). **j** Mean ± s.e. call frequencies in each social context. Alone_B: *Alone_BeforeRET*, Alone_A: *Alone_AfterRET*, Rejection: *During_Rejection* (data collected from 265 sessions of 55 dyads). **k, l** Schematic of the recording (**k**) and mean ± s.e. total call frequencies and the composition of infant call types in the isolation recording (**l**) ($n = 9$, four males and five females). Small dots indicate the value of the total calls in each session. **m** Mean ± s.e. call proportions over the social contexts. Alone_B: *Alone_BeforeRET*, Alone_A: *Alone_AfterRET*, Rejection: *During_Rejection* (data collected from 166 sessions of 35 dyads. Sessions with low-quality vocal recording were excluded). For **a–c**, **e**, **f**, **j**, and **m**, the asterisk and different letters indicate statistical significance at $p < 0.05$ in the GLMM. The dots show the values of each session, and the colors indicate the caregiver.

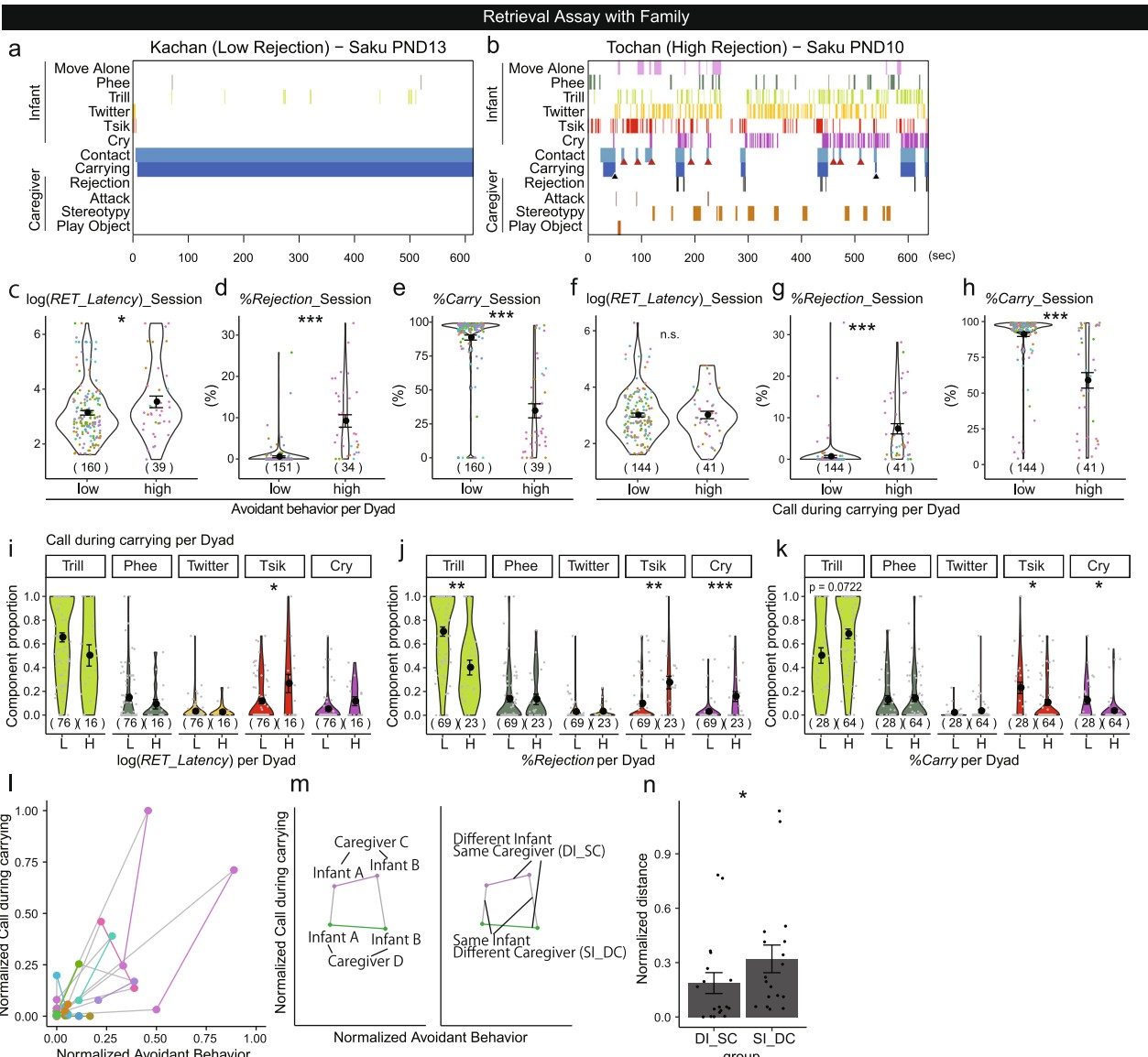

**Fig. 3 | Infants tune their attachment behaviors according to the parenting styles of each caregiver. a** and **b** Representative raster plots of the retrieval assay using an infant Saku (PND 10 and 13), by the mother Kachan (with low *%Rejection*) and the father Tochan (with high *%Rejection*). Black triangles: dismounting without preceding rejection within 9.4 s. Red triangles: refusal to cling when contacted.
**c–h** Violin plots of the parenting parameters (**c**, **f**: *RET_Latency*, **d**, **g**: *%Rejection*, **e**, **h**: *%Carry*) in the groups of dyads with high/low infant avoidance (**c–e**) or high/low infant calls during carrying (**f–h**), defined by the average value (199 sessions of 55 dyads). In **d** and **g**, 185 sessions of 54 dyads as sessions without first retrieval were excluded. **i–k** Violin plots of the proportion of infant call types during carrying in each session in the groups of dyads divided by high (H)/low (L) *RET_Latency* (**i**), *%Rejection* (**j**), and *%Carry* (**k**) at the average value (92 sessions of 33 dyads. Sessions without call during carrying were excluded). **l** Scatter plot of infant calls during

carrying and avoidant behaviors per dyad. The data were normalized to 0 to 1 by min −max normalization (($X$−min($X$))/(max($X$)−min($X$)) for both axes. The same color indicates data from the same caregiver. The dots from the same birth were connected by lines as shown in **m** (36 dyads. Dyads without paired data were excluded, see also **m**, **n**). **m** Schema showing the method to compare the variability of two attachment parameters (**n**) of the same caregiver with two littermate infants (DI_SC, colored segments) and that of the same infant with two caregivers (either parents or older siblings) (SI_DC, gray segments). **n** Mean ± s.e. lengths of DI_SC ($n = 18$ pairs) and SI_DC ($n = 18$ pairs). Wilcoxon rank sum test, $p = 0.0480$. Each dot represents a distance. In **c–k**, the numbers within parentheses are the numbers of the sessions during postnatal weeks 0–3. GLMM, *$p < 0.05$, **$p < 0.01$, ***$p < 0.001$. The black circles and error bars show the mean ± s.e. The dots show the values of each session, and the dot colors in **c–h** indicate the caregiver.

were consistent across litters (Supplementary Fig. 4a), throughout infant development and across two births (Supplementary Fig. 4b). These observations suggest that each parent behaved similarly toward different infants, and each infant behaved according to the caregiver's behavior.

The above observations suggest that there are at least two kinds of atypical attachment behaviors of marmoset infants toward inappropriate caregiving: (i) *avoidant behavior*, namely, dismounting and refusal to cling at contact, exhibited prematurely (i.e., within postnatal weeks 0–3) and voluntarily (i.e., not in *During_Rejection*); and (ii) *calls-while-being-carried*, namely, infant calls during the caregiver's carrying. Indeed, the mappings of

individual dyads for the average caregiving parameters (Supplementary Fig. 4b) and the average atypical attachment behaviors (Supplementary Fig. 4c) impressively resembled each other. The atypical attachment behaviors were much more frequent with the caregivers showing either *%Rejection* or *RET_Latency* over the mean ± 1SD than the rest of the caregivers (Supplementary Fig. 4d).

To examine the relations between parenting styles and infant attachment, we divided dyads into two groups, high/low groups for infant avoidant behavior and calls-while-being-carried, and compared three caregivers' parameters between the groups using the generalized linear

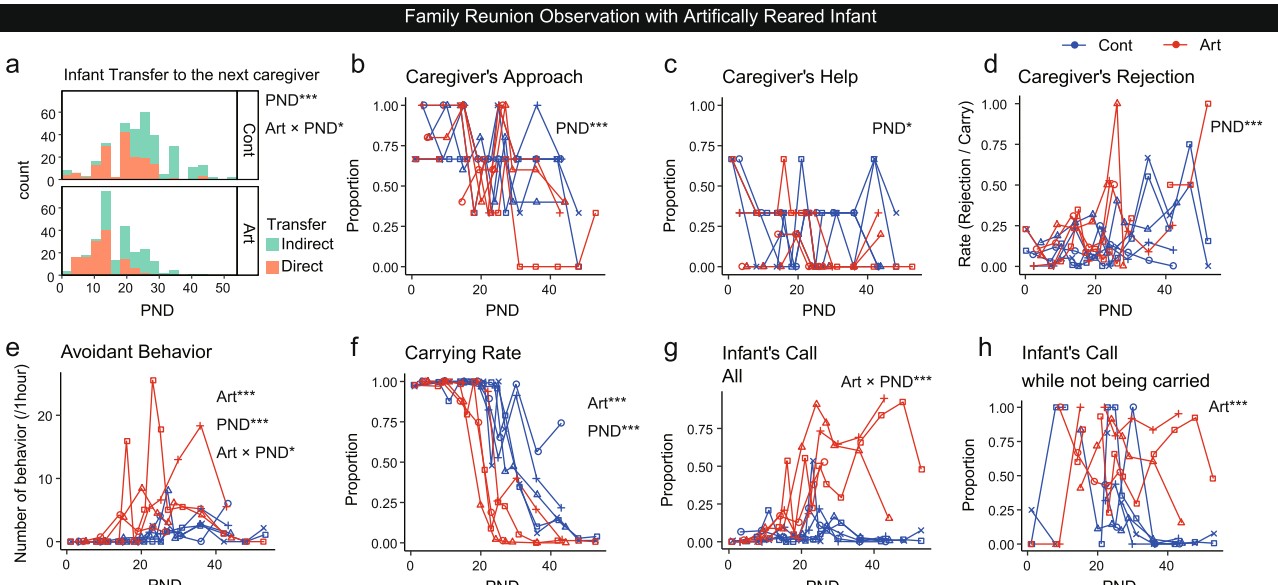

**Fig. 4 | Artificially reared infants showed avoidance and age-disproportionate distress in family settings, even though the caregivers accepted them.**
**a** Histograms of direct and nondirect transfers from one caregiver to another of Art (n = 4 infants, 265 transfers) and Cont infants (n = 5 infants, 324 transfers). Direct transfers decreased with infant PND (z = −7.50, p < 0.001) and declined earlier in Art (z = −2.27, p = 0.0232). **b–h** Caregivers' (**b–d**) and infants' (**e–h**) behaviors during the family reunion. Each marker shape represents each infant. Blue: Cont, red: Art. Numbers of sessions: 96 (**b–g**), 70 (**h**). Numbers of infants; 4 (Art) and 5 (Cont). **b** Proportion of the caregivers that approached the cage entrance upon infant return decreased with infant PND (z = −4.51, p < 0.001). **c** Proportion of caregivers

that attempted to retrieve the returning infant by the forelimb or showed aggressive behaviors toward the experimenter decreased with PND (z = −2.43, p = 0.0149). **d** Caregiver's rejection rate (rejecting bins/carrying bins) decreased with PND (t = 4.992, p < 0.001). **e** Infants' avoidant behaviors were more frequent in Art (z = 4.80, p < 0.001) and increased with PND (z = 4.49, p < 0.001), and the increment was more pronounced in Art (z = −2.39, p = 0.0169). **f** Carrying rate (carried bins/total bins) was lower in Art (t = −5.20, p < 0.001) and decreased with PND (t = −15.29, p < 0.001). **g** Number of bins with infant calls increased with PND in Art (t = 5.30, p < 0.001). **h** Number of bins with infant calls while not being carried was higher in Art (t = 4.45, p < 0.001). *p < 0.05, **p < 0.01, ***p < 0.001 (GLMM).

mixed model (GLMM). The amount of infant avoidant behaviors was correlated with all the caregiving parameters *RET_Latency*, *%Rejection*, and *%Carry* (Fig. 3c–e, Supplementary Fig. 4e, f), even when the caregivers' behavior was equivalent at the moment of the infants' responses. It suggests that infants avoid any caregivers showing insensitivity, intolerance, or scarce carrying. On the other hand, the number of calls-while-being-carried was associated with *%Rejection*, and *%Carry* but not with *RET_Latency* (Fig. 3f–h, Supplementary Fig. 4e, f), suggesting that infants call more while being carried by rejecting (but not insensitive) caregivers. The call type-specific analysis revealed that during carrying by rejecting caregivers, negative calls (tsik and cry) were emitted more frequently, and the positive trill call was emitted less (Fig. 3i–k). Considering the fact that tsik and cry calls were frequently emitted during isolation, and trill calls were emitted while being carried (Fig. 2m), infants might be insecure and exhibit anxiety-like calls when they were carried by rejecting caregivers. These atypical attachment behaviors were positively correlated with each other (r = 0.5167, p < 0.001, Spearman's rank correlation).

These correlations, however, do not infer the causality between care-givers' and infants' behaviors, and it is still possible that the atypical infant behaviors affect caregiving behaviors. We next compared the variabilities of atypical attachment behaviors within the same infant toward different caregivers (SI-DC, gray segments in Fig. 3l, m) vs. variabilities within the different infants toward the same caregiver (DI-SC, colored lines). We found that the variability between different caregivers and the same infant was larger (Fig. 3n). Together with our previous analysis showing the inherent nature of sensitivity and tolerance in each caregiver[37], this result suggests that the caregiver-infant relationship is determined more by the consistent parenting styles of each caregiver than by infant predispositions.

### Family-separated, artificially reared infants as a model of highly insensitive caregiving: Study design
The above-described data strongly suggest that parenting styles causally determine the pattern of infant attachment behaviors. Nevertheless, as these

data are observational, the findings should be confirmed by interventional experiments (see ref. [18]). To this end, we utilized marmoset individuals who were separated from their families in infancy and reared artificially, mod-eling extremely insensitive caregiving. We collected 5 artificially reared infants (Art) from our breeding colony, which were born as triplets or to a mother with a dysfunction of one nipple (detailed in Supplementary Table 6) and could not be maintained in the family even with supplemental formula feeding. Six of their littermates or age- and sex-matched infants were used as controls (Cont). These Art infants were housed individually and reunited with the original family at least 3 h per day for 3–7 days per week in the daytime, except for one (Michael), who was rejected from the family in several trials of reunion and was once attacked (Supplementary Table 6). The subjects were followed from birth and examined along with their development with family reunion observations and infant retrieval assays (Supplementary Fig. 5a).

### Artificially reared infants showed avoidance and age-disproportionate distress in family settings, even though the caregivers accepted them
To assess the effects of artificial rearing on the infant-caregiver interactions within the family, *Family reunion* observations were performed. The observation started by returning the infants to the home cage of their ori-ginal family after daily body weight measurements, and the behaviors of the infants and the family members were coded in every 10-second bin. Our anecdotal observations (Supplementary Fig. 5b) suggested that Art infants emit excessive distress calls while physically avoiding the caregivers. Thus we analyzed the caregivers' and infants' behaviors during the family reunion observations using GLMM, with explanatory variables of Art/Cont and PND (Fig. 4a–h). Typically, when an infant was returned to the home cage after a transient removal (e.g., for body weight examination), the caregivers often approached the returning infants, showed aggression to the experi-menter whose hand grabbed the infant, and tried to reach and draw the infant closer by the forelimb. The proportion of caregivers showing this

https://doi.org/10.1038/s42003-024-05875-6                                                                    **Article**

infant approach in the family declined significantly with infant age but was not different between Art and Cont infants (Fig. 4b). Subsequent caregivers' attempts to retrieve the returned infant (caregivers' help, Fig. 4c) and caregivers' rejection (Fig. 4d) also did not differ between Art and Cont infants. Thus, marmoset caregivers appeared not to discriminate between Art and Cont infants.

On the other hand, infant avoidant behaviors toward caregivers were significantly more frequent among Art infants than among Cont infants after the second postnatal week (Fig. 4e). This might cause a decrease in the total carrying rate of Art infants, starting at a similar time (Fig. 4f).

The duration of each carrying bout, the latency for the first rejection in each carrying bout, and the total number of rejections during each carrying bout declined according to infants' development (Supplementary Fig. 5c–e). Among Art infants, the total number of rejections during carrying declined faster than among Cont infants (Supplementary Fig. 5e), likely due to the reduced tolerance to rejection among Art infants (Supplementary Movie 4). In concordance with this interpretation, the transfer of the infants from one caregiver to the next was different between Art and Cont infants (Fig. 4a). When rejected, Cont infants tended to stick to the caregiver until another caregiver came close and directly moved from the present caregiver to the next (direct transfer) until PND 20. However, Art infants tended to dismount to stay alone before being transferred to the next caregiver (nondirect transfer) after PND 12 (Fig. 4a). Overall, these data suggest that Art infants exhibited compromised approach components of attachment behaviors and increased avoidant behaviors toward caregivers, even though the caregivers behaved similarly toward Art and Cont infants.

Next, vocal behaviors were compared between Art and Cont infants (Fig. 4g, h). In the family reunion observations, the total calls as well as calls-while-not-being-carried were significantly more frequent among Art infants, particularly after PND 30 (Fig. 4g, h, Supplementary Fig. 5b), indicating the age-disproportionate, excessive distress signaling of Art infants (note that in this family setting, calls while being carried could not have been assessed precisely and thus will be described below in the dyadic infant retrieval assays). We also assessed the development of vocalization of one Art infant Michael in the isolated recording condition (Supplementary Fig. 3a–g). Compared with Cont infants (including Cont littermate Cubby), Michael exhibited fewer calls in the first postnatal month, possibly because Michael was used to staying alone, while Cont infants were not. After the second postnatal month, Cont infants reduced the calls, especially cry and tsik calls. However, Michael did not show this trend, again consistent with the delayed independence compared to the family-reared infants.

### Avoidant attachment in Art infants in dyadic infant retrieval assays

We next conducted dyadic retrieval assays with four Art and four Cont infants (Fig. 5, Supplementary Fig. 6, Supplementary Movies 5–7). Consistent with the findings in family reunion, Art infants showed more physical avoidance and distress vocalizations than Cont infants (Fig. 5a, see the legend for details). GLMM analyses revealed that the latencies of caregivers' entry into the cage containing the infant (Supplementary Fig. 6a) and contact with the infant (Supplementary Fig. 6b), which solely depended on caregivers' motivation, were not different between Art and Cont, while the actual retrieval latency was significantly longer for Art (Fig. 5b), possibly because of avoidance of Art infants. In the dyadic situation examined within PND 30, %Carry and %Rejection were not significantly different between Art and Cont (Fig. 5c, d). For infant behavioral parameters, infant avoidant behaviors were more frequent among Art infants (Fig. 5e), as in the family reunion observations. Infant total calls or calls during carrying did not differ between Art and Cont within PND 30 (Fig. 5f, g, Supplementary Fig. 6c) (note that calls-while-not-being-carried were higher after PND 30 in family settings (Fig. 4h)). In the plot of atypical attachment behaviors for Cont and Art infants (Fig. 5h), Cont infants showed few avoidant behaviors and calls-while-being-carried, the latter except for one dyad (Chuck-Nana; note that Chuck was inherently rejecting toward any infants as seen in Supplementary

Fig. 4b-c). In contrast, Art infants often exhibited more avoidant behaviors (Fig. 5h). Furthermore, Cont infants clung mainly to the caregiver's trunk, especially when they became older (Fig. 5i, Supplementary Movie 6). However, Art infants often clung onto the head or neck (Fig. 5i, Supplementary Movie 7). In summary, Art infants showed multiple atypical attachment behaviors, even when caregivers treated them equally to Cont infants in both family reunion and dyadic situations, indicating the pivotal role of social contact during the first two postnatal weeks in the proper development of the infant attachment system.

## Discussion

This study investigated caregiver–infant interactions in common marmosets with high temporal resolutions and elaborated on caregiver-inherent (allo)parenting styles, comprising sensitivity, tolerance and care quantity. These parenting parameters showed significant consistency within each caregiver across infants and experimental paradigms. The sensitivity-tolerance dimensions of each caregiver had a substantial agreement with studies on maternal styles in Old World monkeys[57–61] and humans[4,39,62–64].

This study also detailed the basic features of the infant attachment system in marmosets. Isolated marmoset infants call vigorously and selectively attach to familiar caregivers. Upon contact with the caregiver, they immediately cling on and withhold calls. This clearcut on-off regulation of the infant vocalizations may solicit caregivers' reinforcement learning to retrieve the infant, as loud infant calls should attract predators in the natural habitat. Both signaling and approach behaviors decline after one month of age when the infants gain their locomotor independence. All these features of marmoset infant behaviors are in accordance with the core concept of the infant attachment system defined by Bowlby[4]. Of course, to fully appreciate the similarities and differences of the attachment system in humans and non-human primates, we should not rely only on face validity but consider their underlying mechanisms in each species. The present study aims to present in-depth behavioral information on marmoset attachment system that forms the basis for future neurobiological investigations.

The behavioral responses of infant marmosets toward insensitive or intolerant caregivers parallel further with those of human infants. In humans, an appropriate caregiver is supposed to function as a *safe haven*, to which the infant returns for comfort and support, and as a *secure base*, from which the infant restarts exploration in the environment with a sense of security[65]. If the caregiver inflicts fear or is insensitive to infant distress, infants show insecure attachment, which is classified into avoidant (actively avoid the parent upon reunion in the Strange Situation test), anxious-ambivalent (fails to find comfort in the parent, may appear upset and show tantrums at reunion), and disorganized (may show direct signs of fear and contradictory behaviors simultaneously, thus do not appear to effectively achieve an observable goal)[5–7]. In marmosets, both rejecting and insensitive parenting increase premature avoidant behaviors in infants, even when other carriers are unavailable. These avoidant behaviors may be interpreted as the diminished *safe haven* quality of the caregiver-infant interaction, and may also be regarded as earlier independence earned as an adaptation to an adverse rearing environment (cf. ref. [66]). Another atypical attachment behavior identified in marmoset infants is the insufficient turning-off of vocalizations upon carrying by intolerant caregivers. Infants cling to the intolerant caregiver but emit more negative calls (tsik and cry calls) than trill calls (Fig. 3j) as if they were not carried. Such calls-while-being-carried may indicate anxiety and a decreased *sense of security*. As the *sense of security* obtained in the proximity of the caregiver should be a prerequisite for the subsequent re-departure from the caregiver, calls-while-being-carried may also be relevant to impaired *secure base* behavior.

These atypical or insecure attachment behaviors of family-reared marmoset infants are relation-specific and not a fixed style or predisposition inherent to each infant (Fig. 3a, b, Supplementary Fig. 4a), like insecure attachments of human infants[67]. In other words, marmoset infants are able to avoid a particular caregiver, because they can still rely on other caregivers. Infants of macaque monkeys, dogs, and chicks have been reported to show unaltered or even stronger clinging when their sole caregiver (mother)

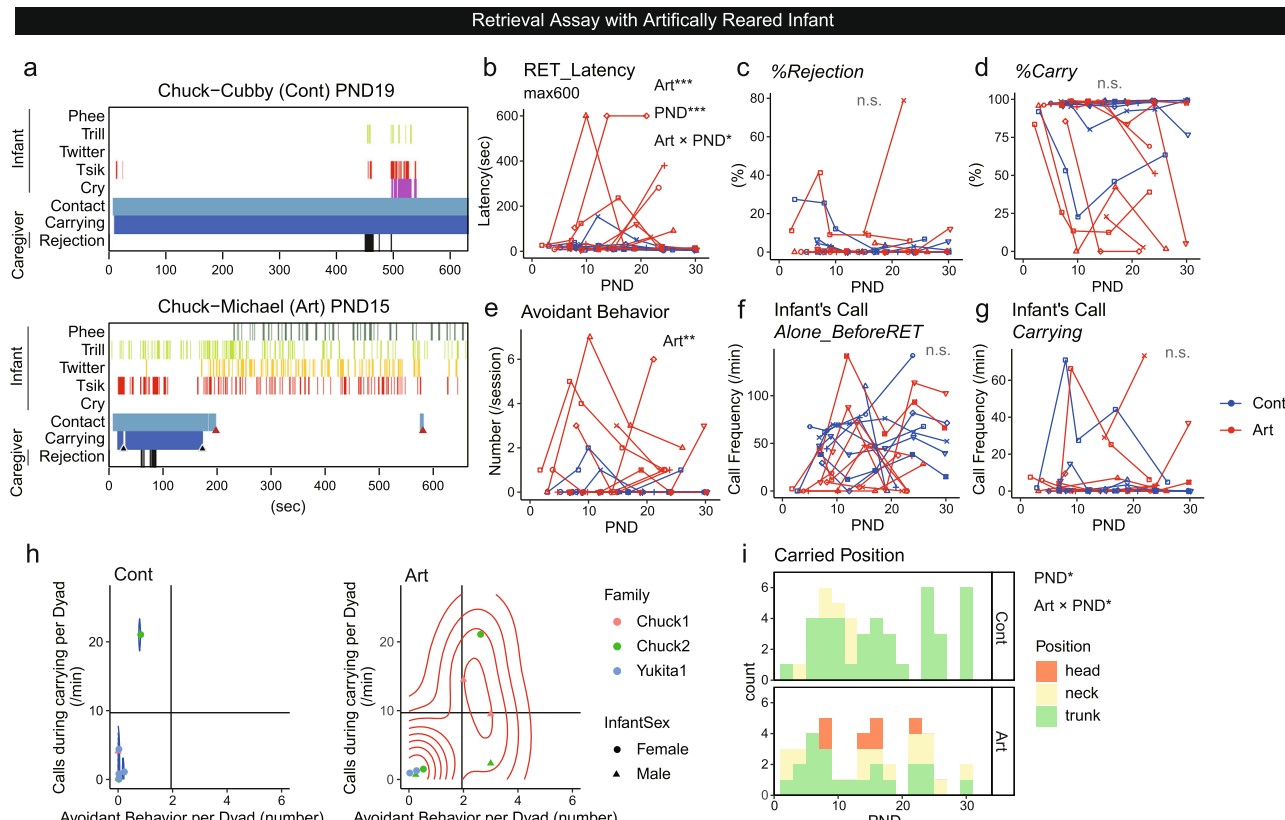

**Fig. 5 | Avoidant attachment in Art infants in dyadic infant retrieval assays. a** Two typical raster plots of the retrieval assay, performed by Cubby (Cont) and Michael (Art) and their father, Chuck. Michael clung on but dismounted from Chuck's body soon after the first retrieval without obvious rejection from Chuck (left black arrowhead) and then was carried again. Michael was rejected once, and a minute later, it dismounted from Chuck without preceding rejection. Chuck made contact again with Michael during 178–190 s, but Michael did not cling onto Chuck (red arrowhead at approximately 190 s, Supplementary Movie 5) and kept calling. The refusal of clinging occurred again at the end of the session (red arrowhead, right). Black arrowhead: voluntary dismounts. Red arrowhead: retrieval refused by the infant. **b–g** Caregiver-infant behaviors and interactions during the retrieval assay. The shape of the markers represents each dyad. Numbers of sessions: 69 (**b, d–f**), 66 (**c, g**; three sessions without retrieval were excluded). Number of dyads: 8 each. **b** Latency of infant retrieval, longer in Art ($t = 3.61$, $p < 0.001$) and the older infants (PND, $t = 3.58$, $p < 0.001$). The coefficient of PND was smaller in Art ($t = -2.54$, $p = 0.0111$). **c** Rejection rate. **d** Percentage of carrying. **e** Infants' avoidant behaviors were more frequent in Art ($z = 3.64$, $p < 0.001$). **f** Frequency of the infant's call before the first retrieval. **g** Frequency of infant calls while being carried. **h** Infant calls during carrying and the number of avoidant behaviors in each dyad, overlaid with the two-dimensional probability density as a contour with 10 bins. Vertical and horizontal lines are the same as in Supplementary Fig. 4d (mean ± 1 s.d in the retrieval assay with family-reared infants). The dot colors indicate the infants. The dot shapes indicate the infant's sex. Number of dyads: 8 each. **i** Infant positioning on the caregiver's body just after the retrieval (Cont: $n = 4$ infants, 47 points, Art: $n = 4$ infants, 40 points). Older Cont infants tended to hold onto the caregivers' trunk ($z = 2.01$, $p = 0.0449$), while Art infants did less so (Art × PND: $z = -2.12$, $p = 0.0342$) and sometimes clung onto the caregiver's head. $*p < 0.05$, $**p < 0.01$, $***p < 0.001$ (GLMM).

behaves abusively (see ref. [68]). This fact may also explain why marmoset infants are resilient when one caregiver is abusive, while macaque infants may receive direct effects from abusive mothers and show delayed independence and excessive anxiety[22].

For the long-term effects of parenting styles on infant development, Baumrind's study is particularly influential by suggesting that sensitive and controlling (may overlap intolerant by use of corporal punishment, see ref. [37]) parenting facilitates independence and self-confidence in preschoolers (older than 3 years 9 months), although the actual data was relatively complicated[38,69]. For infancy, Scott et al.[70] reported that smacking during the first 22 months correlates with behavioral and emotional problems at 4 years of age. Our present data on marmoset infants during the first month (comparable to the first year in humans[71]) appears in line with Scott's study, and we are currently investigating the effects of parenting styles on later behaviors in more detail.

In contrast to the flexibility of the attachment system in family-reared marmosets, artificially reared infants exhibit rigid, fixated avoidance of any caregivers. Moreover, they do not develop age-appropriate autonomy and call excessively while not being carried after one month of age. These calls of Art infants do not function to gain proximity to the caregiver because they simultaneously show contact avoidance. These paradoxical behaviors of the

Art marmosets bear some resemblance to the disorganized attachment[72] or the inhibited type of reactive attachment disorder[73] documented in humans, yet it is too preliminary to propose Art marmosets as a model of human attachment disturbances. Nevertheless, the present results highlight the pivotal role played by the care experienced in the first postnatal month (approximately the first year for human infants) in shaping both approach and signaling components of attachment behaviors, thus forming a basis for dissecting neuromolecular mechanisms of infant attachment and its dysregulations. We are currently studying whether and how Art infants' atypical social and nonsocial behaviors extend into adulthood and which brain areas are involved in separation distress and reunion behaviors in young marmosets.

For vocal development, Fig. 2j–m and Supplementary Fig. 2c demonstrate that infant vocalizations are highly dependent on social contexts (whether they are carried, rejected, or alone/not carried), rather than on infant age during the first 6 weeks. Figure 2l and Supplementary Fig. 3 show that during the postnatal 3 months, the absolute amount of phee calls is stable while those of cry and twitter decrease in the isolated recording condition. These findings collectively suggest that the so-called developmental shift from cry to phee calls, which was proposed in previous studies (for example, ref. [46]) may not be primarily due to the increased ability to

produce phee nor the decreased ability to produce cry, but at least partly due to the developmental decrease in infant need for carrying.

This notion is supported by the previous findings that even young adults emit cry-like calls in a certain context, such as subordination[43,50]. That cry calls can probably be used for begging-like social intent of marmosets in general, and such needs decrease with infant maturation.

The present study offers a caveat in the use of the nomenclature cry for the call originally designated as ngä: Cry is the term used for distress vocalizations mainly in humans, occasionally used as a collective noun that includes different types of vocalizations[74]. Furthermore, the present study has quantitatively demonstrated that marmoset infants use not only cry but also twitter and tsik in their distress vocalizations. Thus, to avoid confusion, it may be preferable to label it as ngä (or like "nghee" to avoid Umlaut), the sound-based label as all the other call types, rather than as cry.

Chronic inhibition of naturalistic parent–infant interactions should disturb both early vocal learning and infant attachment formation; the latter hinders the maturation of autonomy and affect-regulation in primates. Both vocal learning and attachment formation can independently influence vocal patterns, depending on social contexts and developmental stages (e.g., Supplementary Fig. 3). As a result, these data are complex yet coherently understood utilizing the concept of the infant attachment system, which requires proper parental care to be matured[15,18].

We quickly add that the present findings neither contradict nor disregard the importance of vocal learning from adult feedback in marmosets. We also admit that our study is not elaborated enough for fine vocal analyses, as our main focus is on parent-infant relations. The present study instead proposes the infant attachment system as an additional regulator of infant vocalization in various social contexts. Appreciating infant vocalization as a signaling component of the attachment system may help a precise understanding of the development of vocal patterns in infant marmosets, including babbling.

## Methods
### Animals (normative family)
All animal experimentation was approved by the Animal Experiment Judging Committee of RIKEN (equivalent of Institutional Animal Care and Use Committee, IACUC, approval numbers H28-2-210, H30-2-206, W2020-2-027) and was conducted in accordance with the 2011 guideline from the National Research Council of the national academies. Common marmosets were reared at the RIKEN Center for Brain Science in accordance with the institutional guidelines and under veterinarians' supervision. We have complied with all relevant ethical regulations for animal use.

We tested 65 common marmosets (0–8 years old) from 9 families: 8 fathers, 9 mothers, 32 siblings (19 males and 13 females, 3 out of 32 were also included as two fathers and a mother later), and 49 infants (29 males and 20 females, 30 out of 49 were also included as siblings and/or fathers later). Detailed information about the subjects is shown in ref. [37]. They were housed as a family, ranging from parents and two infants (minimum) to parents, two older siblings, two younger siblings, and two infants (maximum). The average number of infants at one birth was 2.21.

One cage was 43 (width) × 66 (height) × 60 (depth) cm. Two or three cages were joined through a square hole (9.6 cm wide × 10.5 cm high) on a side panel or a metal mesh tunnel (75 cm wide × 30.5 cm high × 21 cm deep) placed in front of two cages, depending on the number of family members in accordance with the ethical guideline of RIKEN, to form one single home cage. Each cage contained a food tray, a water faucet, two wooden perches, and a metal mesh loft. Although tactile contact was restricted between families, visual, olfactory, and auditory communication was possible in the colony room. Water and food were supplied ad libitum. The monkeys' food was replenished at approximately 11:30, and supplementary foods, such as a piece of a sponge cake, dried fruits, and lactobacillus preparation, were replenished at approximately 16:00. The photoperiod of the colony room was 12 L:12 D (light period: 8:00–20:00, dark period: 20:00–8:00). The observations and experiments were conducted in the animals' home cages between 8:00 and 17:00. All marmosets

were well habituated to the presence of the experimenters (K.S., S.Y.-N., and a technical staff member) in the colony room to conduct the observations and experiments.

### Evaluation of the caregiver–infant relationship
Caregiver–infant interactions were studied through four structured assays of parental behaviors as described[37]: briefly, (1) instantaneous scan sampling of the family cage, 5 times per day from PND 0 to PND 92 (minimum and maximum data points in the dataset; the same hereinafter), to record the identity of the carrier(s) and the number of infants being carried; (2) continuous 20-min focal observation of the family from PND 0 to 60, to record each family member's caretaking behavior, social behavior, and nonsocial behavior in each 30-s bin; (3) dyadic infant-retrieval assay, starting from 1 to PND 41; and (4) one-to-one food transfer assay from PND 69 to PND 128. In our previous study, all of these data were coded on-site for caregiving behaviors and used to yield a set of parameters as summarized in Supplementary Table 3 (twins are pooled because the individual was not distinguished in the observation)[37]. From this dataset, we performed microanalyses of infant retrieval assays for both caregivers and infants as described below.

### Infant retrieval assay
The dyadic infant retrieval assays, or brief separation–reunion sessions from the infant's viewpoint, were based on the previous marmoset literature[37,75]. The stress loaded onto animals was minimized using their home cage as the testing arena, although some stress was inevitable because of the temporary separation of other family members from the testing area and a brief period (maximum 600 s) of infant isolation.

The infant retrieval assay (Fig. 1a) was conducted from PND 1 to 41 as described[37]. An infant was presented to one of its parents or elder siblings when we returned an infant to its home cage after daily body weight measurement. If more than two caregivers from each family were tested in the same parturition or an infant was a singleton, the infants were separated twice on the same day. The order of the test of multiple family members was counterbalanced. When there were twins in one family, the stimulus infants were alternated. All family members were acclimatized to the tunnel and the wire mesh basket without infants before the test. During the test, the joined home cages were divided into three parts (43 × 66 × 60 cm each) using steel partitions. Two of these cages were connected by a mesh tunnel (75 × 30.5 × 21 cm) and used for the test (Fig. 1a). The caregiver was placed in the left cage before the start of the test. The other family members were placed in the third cage not used for the experiment. During this procedure, the caregiver and the other family members were gently separated by being lured with a piece of sponge cake to minimize the effect of handling on the caregiver's subsequent behavior. Then, the infant was gently taken up from a carrier and placed into the mesh basket (15 cm in diameter × 15 cm high), which contained a gauze-covered electric hand warmer (KIR-SE1S, Sanyo, Osaka, Japan) to maintain the body temperature of the infant during separation. The infant in the mesh basket was placed in the right cage. Opening the shutter of the caregiver's cage permitted the caregiver to access the infant's cage. The behavior of the caregivers and infants before retrieval and 600 s after retrieval, or for 600 s after the opening of the shutter when retrieval was not attempted, was directly observed and recorded using two video cameras (HDR-AS100V, Sony, Tokyo, Japan) as well as a directional microphone (MKH 416, Sennheiser, Hanover, Germany) connected to a linear PCM recorder (DR-60DMKII, Tascam, Tokyo, Japan). The audio was recorded at 24-bit and 96 kHz. The time from the opening of the sliding door to the retrieval of the infant, which was defined as when all of the infant's limbs were in contact with the caregiver's body, was recorded as the retrieval latency. Immediately after successful retrieval, the caregivers' infant-directed behaviors were coded on-site for 600 s with 30-s bins. The session ended if the caregiver did not retrieve the infant for 600 s (or 300 s for the initial 12.6% of these experiments). Twenty-three (2.8%) and 39 (4.8%) sessions ended without retrieval for 300 and 600 s, respectively, among 815 sessions in total.

## Microanalysis of the infant retrieval assay

Using infant retrieval assays recorded by video and a directional microphone[37], manual microanalyses were completed for 286 sessions using 7 families (Supplementary Table 2) and investigated in this study. The video was analyzed with 0.2-s bins for the behaviors listed in Supplementary Table 1 using Solomon Coder (ver. 19.08.02). Infant's and caregiver's vocalizations were detected and classified from the spectrogram of the audio using PRAAT (ver. 6.0.43). For the detailed vocal analyses, we used data from 171 sessions with high-quality vocal recordings.

In the statistical analyses, 21 sessions during the first postnatal week with the caregiver who performed the infant retrieval assay for the first time were regarded as training and excluded from the statistical analysis[37]. The percentage of rejection (*%Rejection*) and infant calls during carrying were calculated in a session with at least one retrieval. The dyadic averages of these parameters were not calculated in an infant–caregiver dyad, Pearl-Mogol, because Mogol did never retrieve Pearl in the three sessions used in the analysis.

Among 164 observed dismounts, 34 dismounts occurred without a preceding rejection. The other 130 dismounts occurred after the rejection with variable intervals ranging from 0.4 to 417.2 s. To divide these events into forced and voluntary dismounts, we performed log-survivorship analysis for rejection-dismount intervals; the 130 dismounts that occurred after rejection are plotted for the intervals from the preceding rejection in a serial order. Then, to divide these events into forced and voluntary dismounts, we performed a segmented regression analysis and found the inflection point at 9.37 s. The rejection-dismount intervals shorter than this threshold were regarded as influenced by the preceding rejection (66.5%, 109 out of 164 dismounts). Twenty-one dismounts with an interval more than the threshold and 34 dismounts without any preceding rejection were regarded as voluntary dismounts (33.5%). Voluntary dismounts were rare for infants younger than PND 28 (23.4%, 25 out of 107 cases), while 30 out of 57 cases and 52.6% of dismounts were voluntary on or after PND 28.

To examine the selectivity of infant attachment, we conducted an additional seven sessions using three infants and two unfamiliar, biologically unrelated adults (one male, Oji, and one female, Hime) with multiple parental experiences. The unfamiliar adults were employed for the experiment when they did not nurture preweaning infants. To minimize the chance of aggressive attacks toward the infants by unfamiliar adults, we selected the adults with comparatively less rejecting as caregivers and monitored their behavior on-site to be able to interrupt the experiments if the adults attacked the infants. In the present study, there was no aggression by the adults exceeding the level of normal rejections. To evaluate the caregivers' behavior, seven sessions with two of their own infants were compared as controls (own infant). For the control of the infants' behavior (own parent), seven sessions with two mothers and two fathers of the infants were used. The sex of caregivers and age of infants in the control sessions were matched.

## Call annotations

Infant and caregiver calls and call types were annotated manually. As previously reported[42,76], infant and adult calls are distinct spectral and temporal characteristics; infant phees are shorter and higher in frequency; each twitter phrase is associated with a downward frequency modulation at the end (twitter-hook). The experimenter identified the characteristic calls produced by the infant or the caregiver when the calls were clearly associated with body movements or mouth opening of either individual using two cameras, then inferred the call identity with this spectrographic pattern for the calls when the body or mouth of the caller was not clearly visible. With a directional microphone, the calls from other cages (inevitable as the rest of the family members were contained in the next cage during the retrieval assay sessions) appeared blurred and low intensity in the spectrogram (Fig. 2h, the phee and twitter calls around the timing of retrieval. Compare the following twitter call by the caregiver). When the source of the call could not have been confidently identified, such an ambiguous call was omitted from the analysis. Using this strategy, the infant calls exhibited suitable interrater reliability (Cohen's $\kappa$ = 0.844 (Cry), 0.527 (Phee), 0.742 (Trill),

0.835 (Tsik), and 0.813 (Twitter), where the chance level $\kappa$ = 0). For the caregivers' vocalizations, we spared detailed analyses for the later study, because our screening analysis in Fig. 1c suggests that the caregivers' physical behaviors such as retrieval and rejection are more immediately influential than caregivers' vocalizations on infant behaviors and calls during the free dyadic interaction, and the main scope of this study is to identify the caregiving influences shaping infant attachment behaviors.

## Artificially reared infants

In our breeding colony, we could often maintain the whole litter of triplets with supplemental feeding but not always, depending on the capacity of milk production of the mother and the total caregiving motivation of the family members. The artificial rearing was performed during our effort to save the infants when the family could not maintain all of them. In this study, we described the artificial rearing of 5 infants, 4 from triplets and one from Chuck's family, of which the mother (Fastener) was unable to nurture two infants at a time because of the dysfunction of one nipple. The details of rearing conditions are described for individual cases in Supplementary Table 6. Briefly, infants were isolated from their families on the day of birth. During isolation, they were kept individually in small cages (22 × 14 × 14 cm or larger according to their growth), given a rolled soft cloth to cling to, and placed in an electric incubator to keep the ambient temperature at 30–37 °C (adjusted to their development). The incubator was placed in the same colony room except for two infants (Senazo and Senako), but not necessarily close to the family cage, and visual, olfactory, and auditory communications with other marmosets were constrained but not completely eliminated. During the daytime, each infant was reunited with the family 2–6 h/day, 3–7 times per week, except for one infant (Michal) who had received aggressive rejections from the family members. Experimenters and care staff handled these Art infants for milk feeding (3 times/day), body weight measurements (once/day), and for experiments. Human baby food, chow soaked in milk, and normal chow after 1.5 months old were also provided. After weaning, each Art infant was put into a normal home cage (42 or 43 (wide) × 66 (high) × 60 (deep) cm) with normal chow and water (ad libitum) and singly housed in the same colony room with their family. The body weight of artificially reared infants tended to be lighter than that of control littermates (*p* = 0.0792, Welch's *t*-test), but it was within the range of variations of the family-reared infants.

The family-reared littermates of the artificially reared subjects were used as control subjects, except for Atako. Because Atako's original family was assigned to another experiment, Atako's experiments were conducted with the Chuck family, which bore two infants one day before Atako's birth. Therefore, Atako's age-matched control was assigned to one infant of Chuck's family. Atako died in an accident at PND27, so no further experiments were conducted.

## Family reunion observations of artificially reared infants

To minimize the separation distress of infants, we reintroduced the Art infants to their family cage during the daytime for 3-7 days per week as described in Supplementary Table 6, and the reunion observations were performed during the initial part of such reunion. Four Art and five Cont infants (Supplementary Table 7) were repeatedly employed for the family reunion observations during PND 1–53. The observations were performed once to three times per week. The observations were conducted in a home cage (size is described in Supplementary Table 7) that contained a mother, a father, and one to three older siblings. First, from the family cage, the family-reared Cont infants were removed and subjected to body-weight measurements. Then, Art and Cont infants were introduced into the family cage in random order. We used the period of their stay in the home cage from the introduction and without interruption by daily cage cleaning for quantitative behavioral analysis, of which length varied between 16 and 119 min (mean ± standard deviation: 57.97 ± 25.93 min), as it turned out that the cage cleaning significantly altered their behavior.

Upon the initiation of the family reunion of the Art infants, the experimenter may solicit the carrying of the Art infants if the family

members did not quickly do so, in order to prevent hypothermia (the ambient temperature of the family cage is lower than that of incubators). If non-carrying of the Art infant is prolonged, or if the family attacked the infant, we terminated the reunion session to save the infant. Otherwise, we did not intervene in the natural interaction of the infants and the family members during the reunion.

The caregivers' reaction when the infant was returned to their home cage was categorized as follows: approached, the caregiver approached the cage entrance where the infant was released by the observer; and helped, the caregiver helped the infant to hold its body or tried to attack the observer's hand holding the infant.

The following infants' conditions and behaviors were manually categorized using *one-zero sampling* in 10 s bins: condition—carried, rejected, attacked, and touched by family members; behaviors—calling. The infants' calling was determined by the sound and oral and/or abdominal movements of the infant using video. When the source of the vocalizations could not be attributed to one infant, because multiple individuals were calling or their movements were unclear, it was recorded as calls by unspecified infant(s) and excluded from the analysis. The avoidant behaviors of the infants were observed continuously.

### Infant retrieval assay with artificially reared infants

Artificially reared infants and their control subjects (artificially reared: $n = 4$, control: $n = 5$, PND 2–30, 69 sessions) were repeatedly employed for the retrieval assays in the home cages of their families (Supplementary Table 8). It was performed twice per week. The detailed procedure is described above. The latency of entering the infant cage, touching the infant, and retrieving the infant was measured. The behaviors of caregivers and infants were analyzed in 0.2 s bins as shown above. The caregiver's body part where the infant was holding was determined when the infant stopped just after a carrying bout started.

### Vocal recordings in isolation

Ten infants (artificially reared male: $n = 1$ (Michael), control: $n = 9$ (four males (Cubby, an infant from Chuck's family, and two infants from Oji's family. The three males were not used in the other experiments of the present report), and five females (Eugenie, Shirayuri, Suisen, Mimosa, Kodemari), 170 sessions, PND 3–104) were used for the vocal recordings in isolation. The recordings were performed once or twice per week. It was conducted in another room separated from the colony rooms. The subject was removed from the home cage and transferred to a plastic cage (28 cm × 44 cm × 20.5 cm) placed in the recording room. The experimenter left the room and recorded the infants' vocalizations using a video recorder and the sound recorder (Sennheiser MKH416-P48U3, TASCAM DR-60DMK II) for 600 s. The vocalizations were manually analyzed on PRAAT.

### Statistics and reproducibility

Statistical analyses were conducted using R software (version 3.6.3)[77]. In the retrieval assays with the normative family, statistical analyses were performed by the generalized linear mixed model (GLMM) with some exceptions. The glmer function of the lme4 package[78] was used for the GLMM. The infant and the carrier were added as random effects with an intercept to control pseudoreplication. The fixed effects included in the models are mentioned in the figure legends. The best models were selected using the dredge function in the MuMIn package[78]. For the correlation matrix, we conducted Spearman's rank correlation and Pearson's product–moment correlation analysis. Correlation coefficients and $p$ values adjusted with Holm's method or Benjamini–Hochberg's method were calculated using the corr.test function in the psych package[79]. For the segmented relationships in regression models, we employed the segmented function in the segmented package[80]. The difference in filial behavior was analyzed using the Wilcoxon rank sum test. The Wilcoxon signed-rank test with continuity correction was conducted for the comparison of infant calls for 10 s after and before the shift of the social contexts.

In the family-reunion observations and the retrieval assays using artificially reared marmosets, statistical analyses were performed by GLMM. The family and the infant nested in the family were added as random effects for family-reunion observations, and the caregiver and the infant were added as random effects for retrieval assays to control pseudoreplication. Random effects included intercepts. PND, artificial rearing, and their interaction were added as a fixed effect in the initial model. The best models were extracted using the dredge function in the MuMIn package[81]. For count data with/without an upper limit, binomial/Poisson distributions were adopted. Latency data were transformed logarithmically and then treated as Gaussian distributions. Other data were treated with normal distributions. For the analyses of the holding site in the retrieval assays, the clmm function in the ordinal package was employed instead of the glmer function in the lme4 package to deal with the ordinal response variable[82].

### Reporting summary

Further information on research design is available in the Nature Portfolio Reporting Summary linked to this article.

### Data availability

All data needed to evaluate the conclusions in the paper are present in the paper and/or the Supplementary Materials. The raw video and audio files can be provided by K.O.K. pending a material transfer agreement, due to ethical restrictions. Requests for these raw data should be submitted to: kurodalab@bio.titech.ac.jp.

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

## Acknowledgements

We thank Drs. Michael Numan and Joji Tsunada for fruitful discussion, Sayaka Shindo and Yumi Ogawa for marmoset breeding, and the RIKEN Center for Brain Science, Research Resources Division for animal care. This research was supported by the RIKEN Center for Brain Science (2014–2022) to K.O.K., the Japan Agency of Medical Research and Development (AMED) under grant numbers JP22dm0207001 to H.O. and JP20dm0107144 to K.O.K., Brain/MINDS project 2014 to K.O.K. and No. 16 dm0207003h0003 to K.N., JSPS KAKENHI grant number JP26893327, JP16K19788, JP20K12587 to K.S., JP19K16901 to S.Y.-N., and JP18KT0036, JP22K19486, and 22H02664 to K.O.K., and Takeda Science Foundation 2023 to K.O.K.

## Author contributions

S.Y.-N. designed and carried out experiments described in Figs. 4, 5, Supplementary Figs. 3, 5, 6, analyzed data described in Figs. 1–5 and Supplementary Figs. 2–6 with help from T.K. and E.M., and produced the tables and figures. A.T. designed and carried out the microanalysis described in Figs. 1–3, and Supplementary Figs. 1, 2, 4 with support from G.E., S.Y.-N., K.S. and K.O.K. K.S. designed and carried out experiments described in Figs. 1–3 and Supplementary Figs. 1, 2, 4 with support from A.S. A.M. carried out a part of the experiments described in Figs. 4, 5, and Supplementary Figs. 5, 6 with support from H.O. K.O.K. conceived of and organized the study with A.T., S.Y.-N. and A.S. and with support from K.M.I., H.T. and K.N. and wrote the manuscript with S.Y.-N. and with contributions from all the authors.

## Competing interests

The authors declare no competing interests.

## Additional information

[1]Laboratory for Affiliative Social Behavior, RIKEN Center for Brain Science, Wako, Japan. [2]Laboratory of Physiology, Department of Basic Veterinary Sciences, Graduate School of Veterinary Medicine, Hokkaido University, Sapporo, Japan. [3]Trinity College Institute of Neuroscience, School of Psychology, Trinity College Dublin, Dublin, Ireland. [4]Department of Psychology and Cognitive Science, University of Trento, Rovereto, TN, Italy. [5]Department of Physiology, Keio University School of Medicine, Shinjuku-ku, Japan. [6]Laboratory for Marmoset Neural Architecture, RIKEN Center for Brain Science, Wako, Japan. [7]Department of Agriculture, Tokyo University of Agriculture and Technology, Fuchu, Japan. [8]Department of Brain & Neurosciences, Tokyo Metropolitan Institute of Medical Science, Setagaya-ku, Japan. [9]Center for the Evolutionary Origins of Human Behavior, Kyoto University, Inuyama, Japan. [10]Department of Psychology, Sophia University, Chiyoda-ku, Japan. [11]Kuroda Laboratory, School of Life Science and Technology, Tokyo Institute of Technology, Yokohama, Japan. [12]Laboratory for Circuit and Behavioral Physiology, RIKEN Center for Brain Science, Wako, Japan. [13]Present address: Planning, Review and Research Institute for Social insurance and Medical program, Chiyoda-ku, Japan. [14]Present address: Neural Circuit Unit, Okinawa Institute Science and Technology Graduate University, Onna, Japan. [15]These authors contributed equally: Saori Yano-Nashimoto, Anna Truzzi, Kazutaka Shinozuka. [16]Deceased: Hironobu Tokuno. ✉e-mail: atsaito@sophia.ac.jp; kurodalab@bio.titech.ac.jp

