## [Peer Review File · Communications Biology]

Anxious about rejection, avoidant of neglect: Infant marmosets tune their attachment based on individual caregiver's parenting styleReviewers' comments:

Reviewer #1 (Remarks to the Author):

The authors present a very detailed study of parent infant behavior. They demonstrate that marmoset infants signal their need by context dependent call use, including a selective attachment behavior dependent on caregiver's parenting style.

1) Major claims of the paper:

The main claim is that "... common marmosets provide an ideal model of human parent-infant relations because, like humans, infant attachment is shared among family caregivers, including parents and older siblings using intricate vocal communications." Abstract L. 58-60, similar at the end of the abstract (L. 70-72).

The authors try to justify this by the sentence that mother is the sole primary caregiver in Old World monkeys, unlike humans (L. 106-07).

Countless studies in Old World monkeys and apes have shown that both, Old World monkeys and apes, are successful models to study parent-infant relations. Unlike marmosets, it is an exception that humans give birth to twins. It is also not the normal case that the human father carries the infant most of the time. This is done by mothers, similar to other nonhuman primates.

This point is not well introduced in the introduction and there is no discussion to which degree this cooperative infant care, due to the fact that marmosets raising twins, provide a better model than other nonhuman primates.

Considering the huge amount of studies on other nonhuman primates I think the introduction should be improved.

2) The authors gave a well-established call type, "cry" a new name: "Infant marmosets emit various kinds of call types, including twitter, tsik, trill, phee, and vhee [originally termed "ngâ" or "cry", here we use "vhee" to avoid unintended anthropomorphic interpretation] ..."

The decision to call a "cry" a "vhee" makes no sense and is totally irritating. I know no publication which ever used "vhee" to label cries. All researchers, including researchers working with marmosets, like Elowson, Snowden, Takahashi or Hage, called a "cry", "cry". This has nothing to do with an anthropomorphic interpretation, because all primates, including humans produce cries. Cries are broad banded sounds (more or less tonal), in their structure totally different from "phees" (a sound in which nearly all energy is focused in a single frequency band). To give them a similar sounding label ("vhee") is only irritating.

The authors should use the normal label ("cry").

3) Developmental shift in vocalizations: "This study demonstrates that infant vocalizations are highly dependent on social contexts rather than on infant age during the first 6 weeks (Fig. 2G, I, S2C). Moreover, our analyses show that the developmental shift from vhee to phee calls in isolated conditions is mainly caused by the rapid decrease of vhee and twitter calls in the first two months, resulting in the relative increase of phee calls, ..." (L. 503-507).

Further: "Both vocal learning and attachment formation can independently influence vocal patterns, ..." (L. 516-17). In the introduction the authors wrote: "Undirected (isolated) vocalizations of infant marmosets undergo progressive changes from infant-specific "cry" and babbling to tonal "phee" calls."

Why is this a developmental shift from "vhee" to "phee"? Twitters also decrease. Obviously infants can produce both call types from the beginning (Fig. 2H).

I also cannot see why this should be show an influence on vocal patterns. The patterns remain the same. Fig. 2 shows that marmosets produce cries and phees from the beginning. Just the frequency to produce these call types change over time.

4) Isolation calls:

The authors wrote that infants produce isolation calls and that infants keep producing isolation calls when they are carried by rejecting caregivers. But it remains unclear whether isolation calls are a specific vocal type or several vocal types together can be seen as isolation calls. Looking at the call frequency in isolation Fig. 2H it seems that the infants produce many different call types. Looking at Fig. 2I it seems that they produce many trills, tsiks and cries during isolation, but not so many phees.

The author should explain which vocal type (types) could be seen as isolation calls and how they come to this conclusion.

5) The authors have a huge number of subjects, but sometimes is it difficult to find out how many subject, dyads are used for the different aspects.

"Detailed analyses of 286 infant retrieval assay sessions were conducted involving 7 families, 25 infants, and 55 infant-caregiver dyads from postnatal day (PND) 1 to 36 (Table S1), ..."

Maybe this can be given in the figure or figure legends. Sometimes the authors gave the number of subjects, sometimes not. Fig, 2H: Does this mean that the call type analysis is based on just 9 subjects? Why?

It would also be helpful to have sometimes not only the p value and which test the authors use (e.g. Fig. 5).

Reviewer #2 (Remarks to the Author):

Yano-Nashimoto and colleagues characterized the social behavior of marmoset infants relative to the care they receive from their family. The main findings are that infants adapt their attachment depending on the caregiver's behavior, and that early life care (isolation vs family reared) influences attachment behaviors similarly to what is seen in humans.

The conclusions are supported by the data, and it is, to my knowledge, the first in depth description of marmoset infant attachment behavior. I do not have major criticisms, however, I think the author should trim the amount of data and figures presented, as well as the result section. I only have minor comments and questions. Also, please note that I did not have access to all supplementary tables (S1 to S4 and S6 to S8) but only table S5.

Results and figures:

1) Several figures do not seem to bring important information to me. I believe that removing some would clarify the message and focus the readers' attention to the most relevant aspects of the study.

- Fig S1 could be removed
- Fig 2 needs to be simplified (some panels can go to supplementary)
- Fig 3 could go entirely to supplementary, furthermore, I think that detailed descriptions of an example session should go to methods or legends (or even be deleted). Presently, it makes the results section tedious to read.
- Similar comments regarding fig 5 and 6, move the examples to supplementary, select only a few panels on which to insist (especially since most of them are non-significantly different between art and cont).
- Fig S4 and S5 bring little information.
- Results from fig S6 could just be mentioned in the text, this does not need an entire figure.

Questions and comments:

2) Please specify to what extent the time threshold to distinguish between forced and voluntary

dismounts correlated with the observed behavior (Fig 2B). Were shorter times particularly associated with forced dismounts?

3) line 803, did the author mean "more" instead of "less"?

4) For figure 1C and fig 4H, it seems that the data did not follow a normal distribution. If so, please use non parametric tests such as Spearman's rank correlation, instead of Pearson, which should only be used when the data are normally distributed.

5) Was there systematically a "good" and an "insensitive/neglecting" caregiver in each couple?

Was there a sex effect in the level of care given to infants?

6) line 354, "The experimenter assisted Senazo in clinging to the mother" This sentence reads as if the experimenter actively helped the baby to cling on the parent. Was this the case? This would be problematic! Did the authors mean "observed"?

7) I understand that vocal interaction is not the focus of this study, but did the authors observed different vocal behavior from the parents exposed to isolated infants? Were there differences in call types/frequency between "good" and "neglecting" parents?

8) The authors draw an interesting parallel between marmoset and human infants' attachment behaviors. Is it known whether crying in human babies is correlated to the care received? It felt like this parallel could be further explicated in the discussion.

Referees' comments and our point-by-point responses

*We used color-coding in the revised main text: **Red** color marks the changes in response to Reviewer #1's comments and **blue** to Reviewer #2's comments. Other changes are marked in **green**.

REVIEWER COMMENTS

Reviewer #1 (Remarks to the Author):

The authors present a very detailed study of parent infant behavior. They demonstrate that marmoset infants signal their need by context dependent call use, including a selective attachment behavior dependent on caregiver's parenting style.

1) Major claims of the paper:

The main claim is that "... common marmosets provide an ideal model of human parent-infant relations because, like humans, infant attachment is shared among family caregivers, including parents and older siblings using intricate vocal communications." Abstract L. 58-60, similar at the end of the abstract (L. 70-72).

The authors try to justify this by the sentence that mother is the sole primary caregiver in Old World monkeys, unlike humans (L. 106-07).

Countless studies in Old World monkeys and apes have shown that both, Old World monkeys and apes, are successful models to study parent-infant relations. Unlike marmosets, it is an exception that humans give birth to twins. It is also not the normal case that the human father carries the infant most of the time. This is done by mothers, similar to other nonhuman primates.

This point is not well introduced in the introduction and there is no discussion to which degree this cooperative infant care, due to the fact that marmosets raising twins, provide a better model than other nonhuman primates.

Considering the huge amount of studies on other nonhuman primates I think the introduction should be improved.

Response: We sincerely appreciate your precious time in assessing this manuscript and pointing out key issues for improvement. Texts were colored in **red** for revisions about your comments.

We totally agree with this comment, and have revised the introduction accordingly, as “For this purpose, rodent infants have been studied extensively **and shown to exhibit many attachment behaviors in common with humans**^{2,12,13}. Still, the attachment of rat or mouse pups are not as selective to a particular individual as humans, presumably because these species may engage in communal nursing¹⁴. **In contrast, the infant attachment of primates is shown to be selective toward the attachment figure (usually the biological mother) and is more profoundly impacted by maternal deprivation and isolation rearing**¹⁵⁻¹⁸ than that of rodents¹⁹⁻²¹. Yet, direct examinations of infant attachment security have been limited to a few studies (see^{22,23}), **especially in those species that the infant attachment is shared among multiple caregivers like humans**^{24,25}.”.

2) The authors gave a well-established call type, “cry” a new name: “Infant marmosets emit various kinds of call types, including twitter, tsik, trill, phee, and vhee [originally termed “ngâ” or “cry”, here we use “vhee” to avoid unintended anthropomorphic interpretation] ...”

The decision to call a “cry” a “vhee” makes no sense and is totally irritating. I know no publication which ever used “vhee” to label cries. All researchers, including researchers working with marmosets, like Elowson, Snowdon, Takahashi or Hage, called a “cry”, “cry”. This has nothing to do with an anthropomorphic interpretation, because all primates, including humans produce cries. Cries are broad banded sounds (more or less tonal), in their structure totally different from “phees” (a sound in which nearly all energy is focused in a single frequency band). To give them a similar sounding label (“vhee”) is only irritating.

The authors should use the normal label (“cry”).

Response: Thank you for raising this issue. According to this comment, we replaced the term “vhee” to “**cry/ngä**”, as “**Infants also emit an additional call type, originally termed as “ngä”^{43,45} and recently as “cry”⁴². Vocal development of infant marmosets has been mostly studied under isolated conditions and are reported to change from infantile “cry/ngä” to tonal “phee” calls, due to the maturation of the vocal apparatus^{46,47} and social learning from vocal feedback from parents^{48,49}. However, it is also reported that infants possess the ability to produce phee right after birth, and retain “cry” calls at 62 postnatal weeks⁵⁰.**”.

We then explained the reason why we avoid simple use of “cry” in more details in the Discussion as “The present study offers a caveat in the use of the nomenclature “cry” for ngä call: “Cry” is the term used for distress vocalizations mainly in humans, occasionally used as a collective noun that includes different types of vocalizations ⁶⁹. Furthermore, the present study has quantitatively demonstrated that marmoset infants use not only cry/ngä but also twitter and tsik in their distress vocalizations. Thus, to avoid confusion, it may be preferable to label it as “ngä” (or like “nghee” to avoid Umlaut), the sound-based label as other call types, rather than as “cry”.”.

3) Developmental shift in vocalizations: “This study demonstrates that infant vocalizations are highly dependent on social contexts rather than on infant age during the first 6 weeks (Fig. 2G, I, S2C). Moreover, our analyses show that the developmental shift from vhee to phee calls in isolated conditions is mainly caused by the rapid decrease of vhee and twitter calls in the first two months, resulting in the relative increase of phee calls, ...” (L. 503-507).

Further: “Both vocal learning and attachment formation can independently influence vocal patterns, ...” (L. 516-17). In the introduction the authors wrote: “Undirected (isolated) vocalizations of infant marmosets undergo progressive changes from infant-specific “cry” and babbling to tonal “phee” calls.”

Why is this a developmental shift from “vhee” to “phee”? Twitters also decrease. Obviously infants can produce both call types from the beginning (Fig. 2H).

I also cannot see why this should be show an influence on vocal patterns. The patterns remain the same. Fig. 2 shows that marmosets produce cries and phees from the beginning. Just the frequency to produce these call types change over time.

Response: Sorry to be unclear, but our understanding and intention is the same as the reviewer’s. The so-called “developmental shift from cry/vhee to phee” (for example, Takahashi et al, 2015) is an apparent phenomenon and the infants can produce both cry/vhee and phee from day 1. What changes during development is the ability to stay alone without panic, rather than the ability to produce phee. To make these point clearer, we revised the sentence as;

Introduction: “Additionally, marmosets' vocal **communication** has attracted considerable attention for its **complexity and human language-like features such as turn-taking and infant “babbling”, or continuous strings of multiple call types which can last for minutes** ^{40,41} ⁴². Marmosets’ **multiple call types show distinct acoustic features and include phee, twitter, tsik, trill, chatter, and Chirp (Table S2)** ^{43,44}. Infants also emit an additional call type, originally termed as “ngä” ^{43,45} and recently as “cry” ⁴². **Vocal development of infant marmosets has been mostly studied under isolated conditions and are reported to change from infantile "cry/ngä" to tonal "phee" calls, due to the maturation of the vocal apparatus** ^{46,47} and social learning from vocal feedback from parents ^{48,49}. **However, it is also reported that infants possess the ability to produce phee right after birth, and retain “cry” calls at 62 postnatal weeks** ⁵⁰. **Moreover, as very young infants are continuously carried by the caregiver and infant calls function to attract parental approach** ⁵¹⁻⁵³, an investigation of infant call development **within intact family settings should be performed (see** ^{42,54})”.

Discussion: “In this study, Fig. 2G, H, and S2C demonstrate that infant vocalizations are highly dependent on social contexts (**whether they are carried, rejected, or Alone/not carried**), rather than on infant age during the first 6 weeks. Fig. 2I and S3 show that during the postnatal 3 months, the absolute amount of phee calls is stable while those of **cry/ngä** and twitter decrease in the isolated recording condition. These findings collectively suggest that the “developmental shift from **cry/ngä** to phee calls”, which was proposed in previous studies (for example, ref ⁴⁶) **may not be primarily due to the increased ability to produce phee nor the decreased ability to produce cry/ngä**, but at least partly due to the developmental decrease in infant need for carrying..”

4) Isolation calls:

The authors wrote that infants produce isolation calls and that infants keep producing isolation calls when they are carried by rejecting caregivers. But it remains unclear whether isolation calls are a specific vocal type or several vocal types together can be seen as isolation calls. Looking at the call frequency in isolation Fig. 2H it seems that the infants produce many different call types. Looking at Fig. 2I it seems that they produce many trills, tsiks and cries during isolation, but not so many phees.

The author should explain which vocal type (types) could be seen as isolation calls and how they come to this conclusion.

Response: Sorry for any confusion, but “isolation call” can mean two different situations in this study. One is during complete isolation in the recording box (Fig. 2i), in which infants cannot see or hear other members. The other is *Alone* (=while not-being-carried, but in the same room with one caregiver) situations within dyadic setting (Fig. 2h). Difference between Fig 2i and 2h indicates that infants properly use different call types according to these distinct conditions.

To be clearer for this important point, we have revised the main text as: “Moreover, infants emitted more phee calls in the completely isolated recording condition (Fig. 2I, S3A-G) than in while-not-being-carried (*Alone*) conditions in the dyadic retrieval assays (compare Fig. 2K and Fig. S2C). This fact implies that infants in the first week of life use phee calls as distant contact calls toward invisible family members as adults do.”

5) The authors have a huge number of subjects, but sometimes is it difficult to find out how many subject, dyads are used for the different aspects.

“Detailed analyses of 286 infant retrieval assay sessions were conducted involving 7 families, 25 infants, and 55 infant-caregiver dyads from postnatal day (PND) 1 to 36 (Table S1), ...”

Maybe this can be given in the figure or figure legends. Sometimes the authors gave the number of subjects, sometimes not. Fig, 2H: Does this mean that the call type analysis is based on just 9 subjects? Why?

Response: We indicate the number of sessions and subjects/dyads in each figure legend.

6) It would also be helpful to have sometimes not only the p value and which test the authors use (e.g. Fig. 5).

Response: We indicate the statistical tests we used in each figure legend.

Reviewer #2 (Remarks to the Author):

Yano-Nashimoto and colleagues characterized the social behavior of marmoset infants relative to the care they receive from their family. The main findings are that infants adapt their attachment depending on the caregiver's behavior, and that early life care (isolation vs family reared) influences attachment behaviors similarly to what is seen in humans.

The conclusions are supported by the data, and it is, to my knowledge, the first in depth description of marmoset infant attachment behavior. I do not have major criticisms, however, I think the author should trim the amount of data and figures presented, as well as the result section. I only have minor comments and questions. Also, please note that I did not have access to all supplementary tables (S1 to S4 and S6 to S8) but only table S5.

Response: We appreciate your precious time for evaluation of this manuscript. The changes made for your comments were color coded by **Blue**.

Results and figures:

1) Several figures do not seem to bring important information to me. I believe that removing some would clarify the message and focus the readers' attention to the most relevant aspects of the study.

- Fig S1 could be removed
- Fig 2 needs to be simplified (some panels can go to supplementary)
- Fig 3 could go entirely to supplementary, furthermore, I think that detailed descriptions of an example session should go to methods or legends (or even be deleted). Presently, it makes the results section tedious to read.
- Similar comments regarding fig 5 and 6, move the examples to supplementary, select only a few panels on which to insist (especially since most of them are non-significantly different between art and cont).
- Fig S4 and S5 bring little information.
- Results from fig S6 could just be mentioned in the text, this does not need an entire figure.

Response: Thank you very much for your effort in evaluating our manuscript. According to this comment, we moved the following main figure panels to the supplementary

material, and moved most of the detailed descriptions of an example session to the relevant legends of the supplemental figures.

(Original Figure numbers)

Fig. 2 I, J

The entire Fig3, except for the left half of the A combined with original Fig. 4 and renamed as Fig. 3.

Fig5a, b

Fig6acdi

Fig. 2 was further simplified by abbreviating the nomenclature of several parameters.

We further deleted the original Fig. S5 and S6 and movie 8 from this manuscript. Fig. S1 is kept to show the overall caregivers' actions toward each infants.

With these modifications, we hope the revised manuscript is better streamlined and concise.

Questions and comments:

2) Please specify to what extent the time threshold to distinguish between forced and voluntary dismounts correlated with the observed behavior (Fig 2B). Were shorter times particularly associated with forced dismounts?

Response: Yes, as described in the legend of Fig. 2B, and in the main text **“Approach” components of the attachment system of infant marmosets**” section.

3) line 803, did the author mean “more” instead of “less”?

Response: Thank you very much for your careful reading. We corrected this mistake.

4) For figure 1C and fig 4H, it seems that the data did not follow a normal distribution. If so, please use non parametric tests such as Spearman's rank correlation, instead of Pearson, which should only be used when the data are

normally distributed.

Response: We have replaced these figures analyzed by Spearman's rank correlation tests. The main conclusions were not affected by this change of statistical method. Minor changes were as follows and marked in the main texts

- p and r values were updated.
- We removed the regression line of original Fig. 4H and instead added the scatter plot (new Fig. I) of infant behaviors in a manner described in new Fig. 4J and K.

5) Was there systematically a “good” and an “insensitive/neglecting” caregiver in each couple? Was there a sex effect in the level of care given to infants?

Response: Yes, each caregiver's parenting style is consistent across infants and multiple births, as described in our previous study (Shinozuka et al, 2022). For example, a father Tochan (Fig. 3A) was consistently rejecting but sensitive (short retrieval latency). Other fathers Beads and Mogol were consistently slow to retrieve, but non-rejective. We now explain this in the result as “These caregiving parameters obtained in the present study were stable in each caregiver toward multiple infants across births, and consistent with their other (allo)parental indices obtained by the intact family observation in our previous study ³⁷ (Table S3) (e.g., %Carry vs. Scan_Carrying rate, Family_Carrying Duration (B)). These observations confirm the existence of a caregiver-inherent (allo)parenting style as demonstrated.”

Each couple may contain one good and one bad parents (e.g., Tochan-Kachan couple, Fig. 3A, S2D) both “good” parents (e.g., Oji-Hime couple, Fig S2E left), both intermediate parents (e.g., Chuck-Fastner couple, Fig. S2E). It is rare that both parents are bad.

6) line 354, “The experimenter assisted Senazo in clinging to the mother” This sentence reads as if the experimenter actively helped the baby to cling on the parent. Was this the case? This would be problematic! Did the authors mean “observed”?

Response: This is due to that these artificial rearing were performed during our effort to save lives of infants that were not sufficiently cared by the family. To minimize the separation distress of infants, we reintroduced the Art infants to their family cage during the daytime for 3-7 days per week as described in Table S6, and the mentioned

observation is the initial part of such reunion. Upon the initiation of family reunion of the Art infant, the experimenter may solicit the carrying of the Art infants if the family members did not quickly do so, in order to prevent hypothermia (the ambient temperature of family cage is lower than that of incubators). If non-carrying of the Art infant is prolonged, or if the family attacked the infant, we terminated the reunion session to save the infant. Otherwise, we did not intervene the natural interaction of the infants and the family members during the reunion.

We added this explanation to the figure legend of Method.

7) I understand that vocal interaction is not the focus of this study, but did the authors observe different vocal behavior from the parents exposed to isolated infants? Were there differences in call types/frequency between “good” and “neglecting” parents?

Response: Thank you for raising this issue. As anecdotally mentioned previously (Pistorio, Epple), adults seldom call back to infant calls even when they approach to the infant. However, we noticed that some “good” caregivers (e.g. the best father OJI in our colony) produce affiliative calls upon retrieval. This issue will be studied in more details in our ongoing study.

8) The authors draw an interesting parallel between marmoset and human infants' attachment behaviors. Is it known whether crying in human babies is correlated to the care received? It felt like this parallel could be further explicated in the discussion.

Response: According to this comment, we added the relevant issues in human infant crying/negative affects in insecure attachment and their relationships with parental behaviors in the Discussion, as

“ The attachment features of infant marmosets toward insensitive or intolerant caregivers exhibit further parallels with those of human infants. In humans, an appropriate caregiver is supposed to function as a *safe haven*, to which the infant returns for comfort and support, and as a *secure base*, from which the infant restarts exploration in the environment with a sense of security ⁶⁴. If the caregiver inflicts fear or is insensitive to infant distress, infants show “insecure” attachment, which is classified into “avoidant”

(actively avoid the parent upon reunion in the Strange Situation test), “anxious-ambivalent” (fails to find comfort in the parent, may appear upset and show tantrums at reunion), and “disorganized” (may show direct signs of fear and contradictory behaviors simultaneously, thus do not appear to effectively achieve an observable goal)⁵⁻⁷.”

“For the long-term effects of parenting styles on infant development, Baumrind’s study is particularly influential by suggesting that “sensitive and controlling (may overlap intolerant by use of corporal punishment, see³⁷)” parenting facilitates independence and self-confidence in preschoolers (older than 3 years 9 months), although the actual data was relatively complicated^{38,69}. For infancy, Scott et al.⁷⁰ reported that smacking during the first 22 months correlates with behavioral and emotional problems at 4 years of age. Our present data on marmoset infants during the first month (comparable to the first year in humans⁷¹) appears in line with Scott’s study, and we are currently investigating the effects of parenting styles on later behaviors in more detail.”

Other changes: (Major changes were colored in green)

1. The abstract was shortened to meet the regulation of Communications Biology.
2. We have corrected “ngâ” to “ngä” in the call name of Epple’s study.

REVIEWERS' COMMENTS:

Reviewer #1 (Remarks to the Author):

The authors improved their manuscript by adding more information regarding subject number, statistical results and theoretical background. However, there are still two open issues that should be resolved before publishing the manuscript.

1) One of my questions was that it would be helpful to make it clearer why Old World monkeys and apes are no good models to study infant attachment, although we have many published studies on parent-infant relations.

As an answer the authors changed the last part of their paragraph: "For this purpose, rodent infants have been studied extensively and shown to exhibit many attachment behaviors in common with humans (2,12,13). Still, the attachment of rat or mouse pups are not as selective to a particular individual as humans, presumably because these species may engage in communal nursing (14). In contrast, the infant attachment of primates is shown to be selective toward the attachment figure (usually the biological mother) and is more profoundly impacted by maternal deprivation and isolation rearing (15-18) than that of rodents (19-21). Yet, direct examinations of infant attachment security have been limited to a few studies (see 22,23), especially in those species that the infant attachment is shared among multiple caregivers like humans (24,25)."

Infant attachment is in many primate species not profoundly impacted by maternal deprivation and isolation rearing. Rhesus macaques (citations) are a special case which is not readily applicable to others primate species. Infant Barbary macaques grew up from the first days on well integrated in their social groups, with many interactions to other females, juveniles and even males. This applies equally to several baboon species and other nonhuman primates. The comparison with rodents doesn't clarify why marmosets are a better model than many other primate species. I think that cannot be explained in this single sentence (see above).

2. The authors replaced the term "vhee" by the term "cry/ngä". The reasoning is more confusing than enlightening. All authors who published acoustic marmoset studies used the term "cry" for this call category. I keep thinking that it would be helpful for readers to have the same nomenclature, because there is no real reason to make such a change. Giving a new name the question remained whether there is a difference between "cry" and "cry/ngä". And their argument that the name "cry" is too anthropomorphic does not really count. The term cry (for such kind of sounds) is used for many species. It is not reserved for cry-like human sounds. In addition, the term "cry" is used anyway ("cry/ngä").

Reviewer #2 (Remarks to the Author):

I am happy with the corrections made by the authors, all my questions have been addressed.

Referees' comments and our point-by-point responses

*We used color-coding in the revised main text: **Red** color marks the changes in response to Reviewer #1's comments. **Green** or by Microsoft Change Tracking system for deletions according to the editor's comments. Other minor changes or corrections due to the English proofreading are not marked.

REVIEWER COMMENTS

Reviewer #1 (Remarks to the Author):

The authors improved their manuscript by adding more information regarding subject number, statistical results and theoretical background. However, there are still two open issues that should be resolved before publishing the manuscript.

1) One of my questions was that it would be helpful to make it clearer why Old World monkeys and apes are no good models to study infant attachment, although we have many published studies on parent-infant relations.

As an answer the authors changed the last part of their paragraph: "For this purpose, rodent infants have been studied extensively and shown to exhibit many attachment behaviors in common with humans (2,12,13). Still, the attachment of rat or mouse pups are not as selective to a particular individual as humans, presumably because these species may engage in communal nursing (14). In contrast, the infant attachment of primates is shown to be selective toward the attachment figure (usually the biological mother) and is more profoundly impacted by maternal deprivation and isolation rearing (15-18) than that of rodents (19-21). Yet, direct examinations of infant attachment security have been limited to a few studies (see 22,23), especially in those species that the infant attachment is shared among multiple caregivers like humans (24,25)."

Infant attachment is in many primate species not profoundly impacted by maternal deprivation and isolation rearing. Rhesus macaques (citations) are a special case which is not readily applicable to others primate species. Infant Barbary macaques grew up from the first days on well integrated in their social groups, with many interactions to other females, juveniles and even males. This

applies equally to several baboon species and other nonhuman primates. The comparison with rodents doesn't clarify why marmosets are a better model than many other primate species. I think that cannot be explained in this single sentence (see above).

Response: We apologize for our poor English writing of this part in the original manuscript, but we do not claim that “Old World monkeys and apes are no good models to study infant attachment”, or marmosets are the better model for all aspects of human attachment compared with other primates. We just meant that marmoset infants have multiple primary caregivers so they have diffused attachments. For example, obviously, marmosets are genetically more distant from humans than old-world monkeys. We revised the Introduction to clarify this point as follows, “**Although these New World monkeys are genetically more distant from humans than Old World monkeys, their cooperative infant care systems and the resulting shared infant attachment present a significant interest due to their similarities with those of humans**”.

We are not sure which study you mentioned by stating that “**Infant attachment is in many primate species not profoundly impacted by maternal deprivation and isolation rearing.**”. Direct investigations of infant attachment behaviors after maternal deprivation or isolation rearing are limited compared with those studying endocrine stress responses or cognitive functions. Yet, profound alterations in attachment behaviors after maternal separation or isolation rearing have been documented in many species other than rhesus macaques, including pigtail macaques (Kaufman, 1967), bonnet macaques (Reite 1989), Capuchins (Byrne, 1999), and great apes (Codner 1984). There are species differences, eg. bonnet macaque infants show less depression compared with pigtail and rhesus infants by receiving allomaternal care. Yet, having interactions with other group members during the neonatal period does not mean that these infants do not form specific attachments to the primary caregiver, as seen in human infants.

2. The authors replaced the term “vhee” by the term “cry/ngä”. The reasoning is more confusing than enlightening. All authors who published acoustic marmoset studies used the term “cry” for this call category. I keep thinking that it would be helpful for readers to have the same nomenclature, because there is no real reason to make such a change. Giving a new name the question remained whether there is a difference between “cry” and “cry/ngä”. And their argument that the name “cry” is too anthropomorphic does not really count. The term cry (for

such kind of sounds) is used for many species. It is not reserved for cry-like human sounds. In addition, the term "cry" is used anyway ("cry/ngä").

Response: We have replaced "cry/ngä" to "cry".

Please accept our gratitude for your precious time and effort in reviewing our manuscript.

Other changes: (Major changes were colored in green)

We included individual points to essentially all figures according to the Editor's comment.